# Represent Your Own Policies: Learning with Policy-extended Value Function Approximator

## Abstract

We study Policy-extended Value Function Approximator (PeVFA) in Reinforcement Learning (RL), which extends conventional value function approximator (VFA) to take as input not only the state (and action) but also an explicit policy representation. Such an extension enables PeVFA to preserve values of multiple policies at the same time and brings an appealing characteristic, i.e., *value generalization among policies*. We formally analyze the value generalization under Generalized Policy Iteration (GPI). From theoretical and empirical lens, we show that generalized value estimates offered by PeVFA may have lower initial approximation error to true values of successive policies, which is expected to improve consecutive value approximation during GPI. Based on above clues, we introduce a new form of GPI with PeVFA which leverages the value generalization along policy improvement path. Moreover, we propose a representation learning framework for RL policy, providing several approaches to learn effective policy embeddings from policy network parameters or state-action pairs. In our experiments, we evaluate the efficacy of value generalization offered by PeVFA and policy representation learning in several OpenAI Gym continuous control tasks. For a representative instance of algorithm implementation, Proximal Policy Optimization (PPO) re-implemented under the paradigm of GPI with PeVFA achieves about 40% performance improvement on its vanilla counterpart in most environments.

## 1 Introduction

Reinforcement Learning (RL) has been widely considered as a promising way to learn optimal policies in many decision-making problems [35, 31, 53, 65, 47, 62, 16]. One fundamental element of RL is value function which defines the long-term evaluation of a policy. With function approximation (e.g., deep neural networks), a value function approximator (VFA) is able to approximate the values of a policy under large and continuous state spaces. As commonly recognized, most RL algorithms can be described as Generalized Policy Iteration (GPI) [55]. As illustrated on the left of Figure 1, at each iteration the VFA is trained to approximate the true values of current policy (i.e., policy evaluation), regarding which the policy is further improved (i.e., policy improvement). The value function approximation error hinders the effectiveness of policy improvement and then the overall optimality of GPI [5, 46]. Unfortunately, such errors are inevitable under function approximation. A large number of samples are usually required to ensure high-quality value estimates, resulting in the sample-inefficiency of deep RL algorithms. Therefore, this raises an urgent need for more efficient value approximation methods [61, 4, 12, 25].

An intuitive idea to improve the efficiency value approximation is to leverage the knowledge on the values of previous encountered policies. However, a conventional VFA usually approximates the values of one policy and values learned from old policies are over-written gradually during

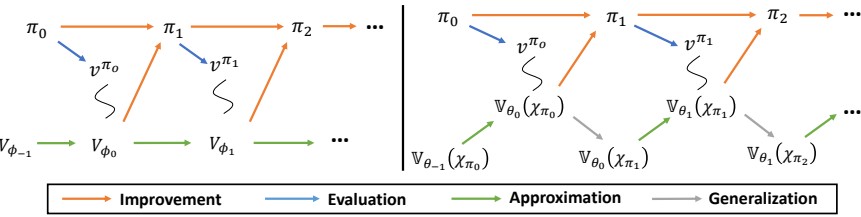

Figure 1: Generalized Policy Iteration (GPI) with function approximation. *Left*: GPI with conventional value function approximator $V_\phi$. *Right*: GPI with PeVFA $\mathbb{V}_\theta(\chi_\pi)$ (Sec. 3) where extra generalization steps exist. The subscripts of policy $\pi$ and value function parameters $\phi, \theta$ denote the iteration number. The squiggle lines represent non-perfect approximation of true values.

the learning process. This means that the previously learned knowledge cannot be preserved and utilized with one conventional VFA. Thus, such limitations prevent the potentials to leverage the previous knowledge for future learning. In this paper, we study Policy-extended Value Function Approximator (PeVFA), which additionally takes an explicit policy representation as input in contrast to conventional VFA. Thanks to the policy representation input, PeVFA is able to approximate values for multiple policies and induces value generalization among policies. We formally analyze the generalization of approximate values among policies in a general form. From both theoretical and empirical lens, we show that the generalized value estimates can be closer to the true values of the successive policy, which can be beneficial to consecutive value approximation along the policy improvement path, called *local generalization*. Based on above clues, we introduce a new form of GPI with PeVFA (the right of Figure 1) that leverages the local generalization to improve the efficiency of consecutive value approximation along the policy improvement path.

One key point of GPI with PeVFA is the representation of policy since it determines how PeVFA generalizes the values. For this, we propose a framework to learn effective low-dimensional embedding of RL policy. We use network parameters or state-action pairs as policy data and encode them into low-dimensional embeddings; then the embeddings are trained to capture the effective information through contrastive learning and policy recovery. Finally, we evaluate the efficacy of GPI with PeVFA and our policy representations. In principle, GPI with PeVFA is general and can be implemented in different ways. As a practical instance, we re-implement Proximal Policy Optimization (PPO) with PeVFA and propose PPO-PeVFA algorithm. Our experimental results on several OpenAI Gym continuous control tasks demonstrate the effectiveness of both value generalization offered by PeVFA and learned policy representations, with an about 40% improvement in average returns achieved by our best variants on standard PPO in most tasks.

We summarize our main contributions below. 1) We study the value generalization among policies induced by PeVFA. From both theoretical and empirical aspects, we shed the light on the situations where the generalization can be beneficial to the learning along policy improvement path. 2) We propose a framework for policy representation learning. To our knowledge, we make the first attempt to learn a low-dimensional embedding of over 10k network parameters for an RL policy. 3) We introduce GPI with PeVFA that leverages the value generalization in a general form. Our experimental results demonstrate the potential of PeVFA in deriving practical and more effective RL algorithms.

## 2 Related Work

**Extensions of Conventional Value Function.** Sutton et al. [56] propose General Value Functions (GVFs) as a general form of knowledge representation of rewards and arbitrary cumulants. Later, conventional value functions are extended to take extra inputs for different purposes of generalization. One notable work is Universal Value Function Approximator (UVFA) [45], which is proposed to generalize values among different goals for goal-conditioned RL. UVFA is further developed in [1, 37, 9] and influences the occurrence of other value function extensions in context-based Meta-RL [43, 29], Hierarchical RL [64] and multiagent RL [19, 14] and etc. Most of the above works study how to generalize the policy or value function among extrinsic factors, i.e., environments, tasks and opponents; while we mainly study the value generalization among policies along policy improvement path, an intrinsic learning process of the agent itself.

**Policy Embedding and Representation.** Although not well studied, representation (or embedding) learning for RL policies is involved in a few works [18, 14, 3]. The most common way to learn a policy representation is to extract from interaction experiences. As a representative, Grover et al. [14] propose learning the representation of opponent policy from interaction trajectories with a generative policy recovery loss and a discriminative triplet loss. These losses are later adopted in [64, 42]. Another straightforward idea is to represent policy parameters. Network Fingerprint [17] is such a differentiable representation that uses the concatenation of the vectors of action distribution outputted by policy network on a set of probing states. The probing state set is co-optimized along with the primary learning objective, which can be non-trivial especially when the dimensionality of the set is high. Besides, some early attempts in learning low-dimensional embedding of policy parameters are studies in Evolutionary Algorithms [13, 44], mainly with the help of VAE [23]. Our work introduce a learning framework of policy representation including both above two perspectives.

**PVN and PVFs.** Recently, several works study the generalization among policy space. Harb et al. [17] propose Policy Evaluation Network (PVN) to directly approximate the distribution of policy $\pi$'s objective function $J(\pi) = \mathbb{E}_{\rho_0}[v^\pi(s_0)]$ with initial state $s_0 \sim \rho_0$. PVN takes as input Network Fingerprint (mentioned above) of policy network. After training on a pre-collected set of policies, a random initialized policy can be optimized in a zero-shot manner with the policy **g**radients of PVN by backpropagting **t**hrough the differentiable **p**olicy **i**nput. We call such gradients *GTPI* for short below. Similar ideas are later integrated with task-specific context learning in multi-task RL [42], leveraging the generalization among policies and tasks for fast policy adaptation on new tasks. In PVN [17], as an early attempt, the generalization among policies is studied with small policy network and simple tasks; besides, the most regular online learning setting is not studied. Concurrent to our work, Faccio and Schmidhuber [10] propose a class of Parameter-based Value Functions (PVFs) that take vectorized policy parameters as inputs. Based on PVFs, new policy gradient algorithms are introduced in the form of a combination of conventional policy gradients and GTPI (i.e., by backpropagating through policy parameters in PVFs). Except for zero-shot policy optimization as conducted in PVN, PVFs are also evaluated for online policy learning. Due to directly taking parameters as input, PVFs suffer from the curse of dimensionality when the number of parameters is high. Besides, GTPI can be non-trivial to rein since policy parameter space are complex and extrapolation generalization error can be large when the value function is only trained on finite policies (usually much fewer than state-action samples) thus further resulting in erroneous policy gradients.

Our work differs with PVFs from several aspects. First, we make use of learned policy representation rather than policy network parameters. Second, we do not resort to GTPI for the policy update in our algorithms but focus on utilizing value generalization for more efficient value estimation in GPI. Furthermore, we shed the light on two important problems — how value generalization among policies can happen formally and whether it is beneficial to learning or not — which are neglected in in previous works from both theoretical and empirical lens.

## 3 Policy-extended Value Function Approximator

In this section, we propose Policy-extended Value Function Approximator (PeVFA), an extension of conventional VFA that explicitly takes as input a policy representation. First, we introduce the formulation (Sec. 3.1), then we study value generalization among policies theoretically (Sec. 3.2) along with some empirical evidences (Sec. 3.3). Finally, we derive a new form of GPI (Sec. 3.4).

### 3.1 Formulation

Consider a Markov Decision Process (MDP) defined as $\langle \mathcal{S}, \mathcal{A}, r, \mathcal{P}, \gamma \rangle$ where $\mathcal{S}$ is the state space, $\mathcal{A}$ is the action space, $r$ is the (bounded) reward function, $\mathcal{P}$ is the transition function and $\gamma \in [0, 1)$ is the discount factor. A policy $\pi \in P(\mathcal{A})^{|S|}$ defines the distribution over all actions for each state. The goal of an RL agent is to find an optimal policy $\pi^*$ that maximizes the expected long-term discounted return. The state-value function $v^\pi(s)$ is defined as the expected discounted return obtained through following the policy $\pi$ from a state $s$: $v^\pi(s) = \mathbb{E}_\pi \left[ \sum_{t=0}^\infty \gamma^t r_{t+1} | s_0 = s \right]$ for where $r_{t+1} = r(s_t, a_t)$. We use $V^\pi$ to denote the vectorized form of value function.

In a general form, we define *policy-extended value function* $\mathbb{V} : \mathcal{S} \times \Pi \to \mathbb{R}$ over state and policy space: $\mathbb{V}(s, \pi) = v^\pi(s)$ for all $s \in \mathcal{S}$ and $\pi \in \Pi$. In this paper, we focus on $\mathbb{V}(s, \pi)$ and policy-extended action-value function $\mathbb{Q}(s, a, \pi)$ can be obtained similarly. We use $\mathbb{V}(\pi)$ to denote the value

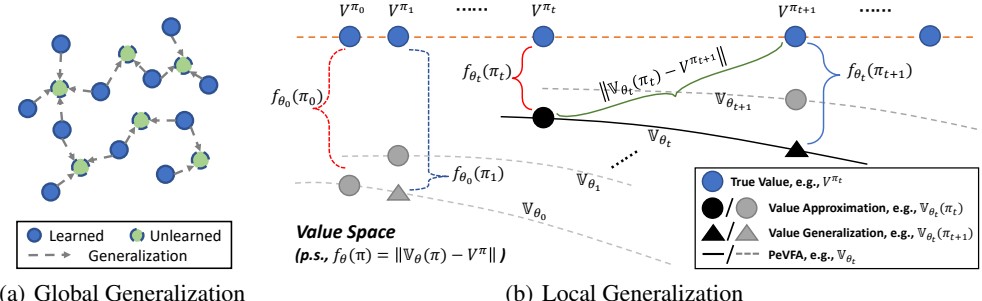

| (a) Global Generalization | (b) Local Generalization |

Figure 2: Illustrations of value generalization among policies of PeVFA. Each circle denotes value function (estimate) of a policy. (a) *Global Generalization*: values learned from known policies can be generalized to unknown policies. (b) *Local Generalization*: values of previous policies (e.g., $\pi_t$) can be generalized to successive policies (e.g., $\pi_{t+1}$) along policy improvement path.

vector for all states in the following. The key point is that PeVFA $\mathbb{V}$ is able to preserve the values of multiple policies. With function approximation, a PeVFA is expected to approximate the values of policies among policy space, i.e., $\{V^\pi\}_{\pi \in \Pi}$ and then enable value generalization among policies.

Formally, given a function $g : \Pi \to \mathcal{X} \subseteq \mathbb{R}^n$ that maps any policy $\pi$ to an $n$-dimensional representation $\chi_\pi = g(\pi) \in \mathcal{X}$, a PeVFA $\mathbb{V}_\theta$ with parameter $\theta \in \Theta$ is to minimize the approximation error over all possible states and policies generally:

$$F_{\mu,p,\rho}(\theta, g, \Pi) = \sum_{\pi \in \Pi} \mu(\pi) \|\mathbb{V}_\theta(\chi_\pi) - V^\pi\|_{p,\rho} \, , \tag{1}$$

where $\mu$, $\rho$ are distributions over policies and states respectively, $\|f\|_{p,\rho} = (\int_s \rho(\mathrm{d}s)|f(s)|^p)^{1/p}$ is $\rho$-weighted $L_p$-norm [26, 46] for any $f : \mathcal{S} \to \mathbb{R}$. The policy distribution $\mu$ of interest depends on the scenario where value generalization is considered. As illustrated in Figure 2, we provide two value generalization scenarios. In the global generalization scenario, a uniform distribution over known policy set may be considered with a general purpose of value generalization for unknown policies. For the specific local generalization scenario along policy improvement path during GPI, a sophisticated distribution that adaptively weights recent policies more during the learning process may be more suitable in this case. In the following, we care more about the local generalization scenario and use uniform state distribution $\rho$ and $L_2$-norm for demonstration. The subscripts are omitted and we use $\| \cdot \|$ for clarity.

## 3.2 Theoretical Analysis on Value Generalization among Policies

In this part, we theoretically analyze the value generalization among policies induced by PeVFA. We start from a two-policy case and study whether the value approximation learned for one policy can be generalized to the other one. Later, we study the local generalization scenario (Figure 2(b)) and shed the light on the superiority of PeVFA for GPI. All the proofs are provided in Appendix A.

For the convenience of demonstration, we use an identical policy representation function, i.e., $\chi_\pi = \pi$, and define the approximation loss of PeVFA $\mathbb{V}_\theta$ for any policy $\pi \in \Pi$ as $f_\theta(\pi) = \|\mathbb{V}_\theta(\pi) - V^\pi\| \geq 0$. We use the following definitions for a formal description of value approximation process with PeVFA and local property of loss function $f_\theta$ that influences generalization [40, 63] respectively:

**Definition 1** ($\pi$-**Value Approximation**) *We define a value approximation process $\mathscr{P}_\pi : \Theta \to \Theta$ with PeVFA as a $\gamma$-contraction mapping on the approximation loss for policy $\pi$, i.e., for $\hat{\theta} = \mathscr{P}_\pi(\theta)$, we have $f_{\hat{\theta}}(\pi) \leq \gamma f_\theta(\pi)$ where $\gamma \in [0, 1)$.*

**Definition 2** ($L$-**Continuity**) *We call $f_\theta$ is $L$-continuous at policy $\pi$ if $f_\theta$ is Lipschitz continuous at $\pi$ with a constant $L \in [0, \infty)$, i.e., $|f_\theta(\pi) - f_\theta(\pi')| \leq L \cdot d(\pi, \pi')$ for $\pi' \in \Pi$ with some distance metric $d$ for policy space $\Pi$.*

With Definition 1, the consecutive value approximation for the policies along policy improvement path during GPI can be described as: $\theta_{-1} \xrightarrow{\mathscr{P}_{\pi_0}} \theta_0 \xrightarrow{\mathscr{P}_{\pi_1}} \theta_1 \xrightarrow{\mathscr{P}_{\pi_2}} \dots$, as the green arrows illustrated in Figure 1. One may refer to Appendix A.1 for a discussion on the rationality of the two definitions.

To start our analysis, we first study the generalized value approximation loss in a two-policy case where only the value of policy $\pi_1$ is approximated by PeVFA as below:

**Lemma 1** *For $\theta \xrightarrow{\mathscr{P}_{\pi_1}} \hat{\theta}$, if $f_{\hat{\theta}}$ is $\hat{L}$-continuous at $\pi_1$ and $f_\theta(\pi_1) \leq f_\theta(\pi_2)$, we have: $f_{\hat{\theta}}(\pi_2) \leq \gamma f_\theta(\pi_2) + \mathcal{M}(\pi_1, \pi_2, \hat{L})$, where $\mathcal{M}(\pi_1, \pi_2, \hat{L}) = \hat{L} \cdot d(\pi_1, \pi_2)$.*

**Corollary 1** *$\mathscr{P}_{\pi_1}$ is $\gamma_g$-contraction ($\gamma_g \in [0, 1)$) for $\pi_2$ when $f_\theta(\pi_2) > \frac{\hat{L} \cdot d(\pi_1, \pi_2)}{1 - \gamma}$.*

Lemma 1 shows that the post-$\mathscr{P}_{\pi_1}$ approximation loss for $\pi_2$ is upper bounded by a generalized contraction of prior loss plus a locality margin term $\mathcal{M}$ which is related to $\pi_1$, $\pi_2$ and the locality property of $f_{\hat{\theta}}$. In general, the form of $\mathcal{M}$ depends on the local property assumed. Some higher-order variants are provided in Appendix A.2. For a step further, Corollary 1 reveals the condition where a contraction on value approximation loss for $\pi_2$ is achieved when PeVFA is only trained to approximate the values of $\pi_1$. Concretely, such a condition is apt to reach with tighter contraction for policy $\pi_1$ is, closer two policies, or smoother approximation loss function $f_{\hat{\theta}}$.

Then we consider the local generalization scenario as illustrated in Figure 2(b). For any iteration $t$ of GPI, the values of current policy $\pi_t$ are approximated by PeVFA, followed by a improved policy $\pi_{t+1}$ whose values are to be approximated in the next iteration. The value generalization from each $\pi_t$ and $\pi_{t+1}$ can be similarly considered as the two-policy case. In addition to the former results, we shed the light on the value generalization loss of PeVFA along policy improvement path below:

**Lemma 2** *For $\theta_{-1} \xrightarrow{\mathscr{P}_{\pi_0}} \theta_0 \xrightarrow{\mathscr{P}_{\pi_1}} \theta_1 \xrightarrow{\mathscr{P}_{\pi_2}} \ldots$ with $\gamma_t$ for each $\mathscr{P}_{\pi_t}$, if $f_{\theta_t}$ is $\hat{L}_t$-continuous at $\pi_t$ for any $t \geq 0$, we have $f_{\theta_t}(\pi_{t+1}) \leq \gamma_t f_{\theta_{t-1}}(\pi_t) + \mathcal{M}_t$, where $\mathcal{M}_t = L_t \cdot d(\pi_t, \pi_{t+1})$.*

**Corollary 2** *By induction, we have $f_{\theta_t}(\pi_{t+1}) \leq \prod_{i=0}^{t} \gamma_t f_{\theta_{-1}}(\pi_0) + \sum_{i=0}^{t-1} \prod_{j=i+1}^{t} \gamma_j \mathcal{M}_i + \mathcal{M}_t$.*

The above results indicate that the value generalization loss can be recursively bounded and has a upper bound formed by a repeated contraction on initial loss plus the accumulation of locality margins induced from each local generalization. An infinity-case discussion for Corollary 2 is in Appendix A.5. The next question is whether PeVFA with value generalization among policies is preferable to the conventional VFA. To this end, we introduce a desirable condition which reveals the superiority of PeVFA during consecutive value approximation along the policy improvement path:

**Theorem 1** *During $\theta_{-1} \xrightarrow{\mathscr{P}_{\pi_0}} \theta_0 \xrightarrow{\mathscr{P}_{\pi_1}} \theta_1 \xrightarrow{\mathscr{P}_{\pi_2}} \ldots$, for any $t \geq 0$, if $f_{\theta_t}(\pi_t) + f_{\theta_t}(\pi_{t+1}) \leq \|V^{\pi_t} - V^{\pi_{t+1}}\|$, then $f_{\theta_t}(\pi_{t+1}) \leq \|\mathbb{V}_{\theta_t}(\pi_t) - V^{\pi_{t+1}}\|$.*

Theorem 1 shows that the generalized value estimates $\mathbb{V}_{\theta_t}(\pi_{t+1})$ can be closer to the true values of policy $\pi_{t+1}$ than $\mathbb{V}_{\theta_t}(\pi_t)$. Note that $\mathbb{V}_{\theta_t}(\pi_t)$ is the value approximation for $\pi_t$ which is equivalent to the counterpart $V_{\phi_t}$ for a conventional VFA as value generalization among policies does not exist. To consecutive value approximation along policy improvement path, this means that the value generalization of PeVFA has the potential to offer closer start points at each iteration. If such closer start points can often exist, we expect PeVFA to be preferable to conventional VFA since value approximation can be more efficient with PeVFA and it in turn facilitates the overall GPI process.

However, the condition in Theorem 1 is not necessarily met in practice. Intuitively, it depends on the locality margins that may be related to function family and optimization method of PeVFA, as well as the scale of policy improvement. We leave these further theoretical investigations for future work. Instead, we empirically examine the existence of such desirable generalizations in the following.

### 3.3 Empirical Evidences

We empirically investigate the value generalization of PeVFA with didactic environments. In this section, PeVFA $\mathbb{V}_\theta$ is parameterized by neural network and we use the concatenation of all weights and biases of the policy network as a straightforward representation $\chi_\pi$ for each policy, called *Raw Policy Representation (RPR)*. Experimental details are provided in Appendix B.

First, we demonstrate the global generalization (illustrated in Figure 2(a)) in a continuous 2D Point Walker environment. We build the policy set $\Pi$ with synthetic policies, each of which is a randomly initialized 2-layer *tanh*-activated neural network with 2 units for each layer. The size of $\Pi$ is 20k and the behavioral diversity of synthetic policies is verified (see Figure 7(b) in Appendix). We divide $\Pi$ into training set (i.e., known policies $\Pi_0$) and testing set (i.e., unseen policies $\Pi_1$). We rollout the

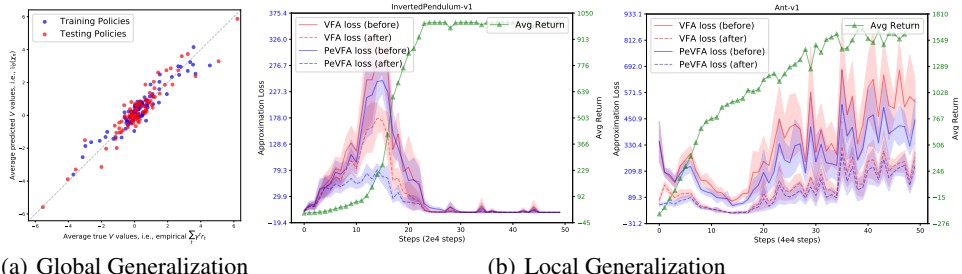

(a) Global Generalization          (b) Local Generalization

Figure 3: Empirical evidences of two kinds of generalization of PeVFA. (a) *Global generalization*: PeVFA shows comparable value estimation performance on testing policy set (red) after learning on training policy set (blue). (b) *Local generalization*: PeVFA ($\mathbb{V}_\theta(\chi_\pi)$) shows lower losses than conventional VFA ($V_\phi$) before and after the value approximation training for successive policies along policy improvement path. In (b), the left axis is for approximation loss (lower is better) and the right axis is for average return as a reference of the policy learning process (green curve).

policies in the environment to collect trajectories, based on which we perform value approximation training. Our results show that a PeVFA trained on $\Pi_0$ achieves reasonable generalization performance when evaluating on $\Pi_1$. The average losses on training and testing set are 1.782 and 2.071 over 6 trials. Figure 3(a) shows the value predictions for policies from training and testing set (100 for each).

Next, we investigate the value generalization along policy improvement path, i.e., local generalization as in Figure 2(b). We use a 2-layer 8-unit policy network trained by standard PPO algorithm [50] in MuJoCo continuous control tasks. Parallel to the conventional value network $V_\phi(s)$ (i.e., VFA) in PPO, we set a PeVFA network $\mathbb{V}_\theta(s, \chi_\pi)$ as a reference for the comparison on value approximation loss. Compared to $V_\phi$, PeVFA $\mathbb{V}_\theta(s, \chi_\pi)$ takes RPR as input and approximates the values of all historical policies ($\{\pi_i\}_{i=0}^t$) in addition. We compare the value approximation losses of $V_\phi$ (red) and $\mathbb{V}_\theta$ (blue) before (solid) and after (dashed) updating with on-policy samples collected by the improved policy $\pi_{t+1}$ at each iteration. Figure 3(b) shows the results for InvertedPendulum-v1 and Ant-v1. Results for all 7 MuJoCo tasks can be found in Appendix B.2. By comparing approximation losses before updating (red and blue solid curves), we can observe that the approximation loss of $\mathbb{V}_{\theta_t}(\chi_{\pi_{t+1}})$ is almost consistently lower than that of $V_{\phi_t}$. This means that the generalized value estimates offered by PeVFA are usually closer to the true values of $\pi_{t+1}$, demonstrating the consequence arrived in Theorem 1. For the dashed curves, it shows that PeVFA $\mathbb{V}_{\theta_{t+1}}(\chi_{\pi_{t+1}})$ can achieve lower approximation loss for $\pi_{t+1}$ than conventional VFA $V_{\phi_{t+1}}$ after the same number of training with the same on-policy samples. The empirical evidence above indicates that PeVFA can be preferable to the conventional VFA for consecutive value approximation. The generalized value estimates along policy improvement path have the potential to expedite the process of GPI.

### 3.4 Reinforcement Learning with PeVFA

Based on the results above, we expect to leverage the value generalization of PeVFA to facilitate RL. In Algorithm 1, we propose a general description of RL algorithm under the paradigm of GPI with PeVFA. For each iteration, the interaction experiences of current policy and the policy

---

**Algorithm 1** RL under the paradigm of GPI with PeVFA ($\mathbb{V}(s, \chi_\pi)$ is used for demonstration)

1: Initialize policy $\pi_0$, policy representation model $g$, PeVFA $\mathbb{V}_{-1}$ and experience buffer $\mathcal{D}$
2: **for** iteration $t = 0, 1, \ldots$ **do**
3:      Rollout policy $\pi_t$ in the environment and obtain $k$ trajectories $\mathcal{T}_t = \{\tau_i\}_{i=0}^k$
4:      Get representation $\chi_{\pi_t} = g(\pi)$ for policy $\pi_t$ and add experiences $(\chi_{\pi_t}, \mathcal{T}_t)$ in buffer $\mathcal{D}$
5:      **if** $t \% M = 0$ **then**
6:          Update PeVFA $\mathbb{V}_{t-1}(s, \chi_{\pi_i})$ for previous policies with data $\{(\chi_{\pi_i}, \mathcal{T}_i)\}_{i=0}^{t-1}$
7:          Update policy representation model $g$, e.g., with approaches provided in Sec. 4
8:      **end if**
9:      Update PeVFA $\mathbb{V}_{t-1}(s, \chi_{\pi_t})$ for current policy $\chi_{\pi_t}$ and set $\mathbb{V}_t \longleftarrow \mathbb{V}_{t-1}$
10:      Update $\pi_t$ w.r.t $\mathbb{V}_t(s, \chi_{\pi_t})$ by policy improvement algorithm and set $\pi_{t+1} \longleftarrow \pi_t$
11: **end for**

---

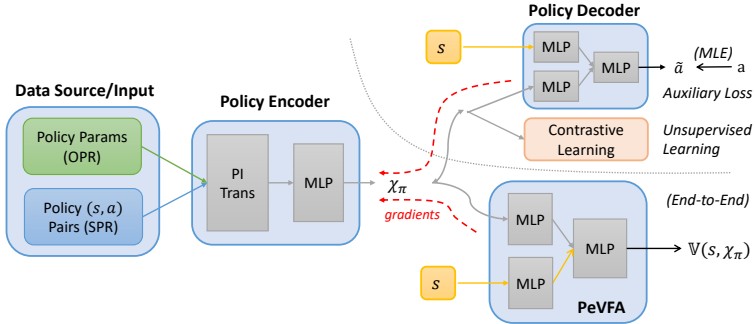

Figure 4: The framework of policy representation training. Policy network parameters used for OPR or policy state-action pairs used for SPR are fed into policy encoder with permutation-invariant (PI) transformations followed by an MLP, producing the representation $\chi_\pi$. Afterwards, $\chi_\pi$ can be trained by gradients from the value approximation loss of PeVFA (i.e., End-to-End), as well as (optionally) the auxiliary loss of policy recovery or the contrastive learning (i.e., InfoNCE) loss.

representation are stored in a buffer (line 3-4). At an interval of $M$ iterations, PeVFA is trained via value approximation for previous policies with the stored data and the policy representation model is updated according to the method used (line 5-8). This part is unique to PeVFA for preservation and generalization of knowledge of historical policies. Next, value approximation for current policy is performed with PeVFA (line 9). A key difference here is that the generalized value estimates (i.e., $\mathbb{V}_{t-1}(\chi_{\pi_t})$) are used as start points. Afterwards, a successive policy is obtained from typical policy improvement (line 10). Algorithm 1 can be implemented in different ways and we propose an instance implemented based on PPO [50] in our experiments later. In the next section, we introduce our methods for policy representation learning.

# 4    Policy Representation Learning

To derive practical deep RL algorithms, one key point is policy representation, i.e., a low-dimensional embedding of RL policy. Intuitively, policy representation influences the approximation and generalization of PeVFA. Thus, it is of interest to find an effective policy representation based on which the superiority of PeVFA can be leveraged to improve RL algorithms. To our knowledge, policy representation is not well studied and it remains unclear on how to obtain an effective representation for an RL policy in a general case in practice. In previous section, we demonstrate the effectiveness of using policy parameters as a naive representation when policy network is small, called RPR. However, a usual policy network may have large number of parameters, thus making it inefficient and even irrational to use RPR for approximation and generalization [17, 10]. More generally, policy parameters of the policy we wish to represent may not be accessible.

To this end, we propose a general framework of policy representation learning as illustrated in Figure 4. The first thing to consider is data source, i.e., from which we can extract the information for an effective policy representation. Recall that the policy is a distribution over state and action space of high dimensionality. The features of such a distribution is not directly available. Therefore, we consider two kinds of data source below that indirectly contains the information of policies: 1) *Surface Policy Representation (SPR)*: The first data source is state-action pairs (or trajectories [14]), since they reflect how policy may behave under such states. This data source is general since no explicit form of policy is assumed. In a geometric view, learning policy representation from state-action pairs can be viewed as capturing the features of policy via scattering sample points on the curved surface of policy distribution. 2) *Origin Policy Representation (OPR)*: The other data source is parameters of policy since they determine the underlying form of policy distribution. Such a data source is often available during the learning process of deep RL algorithms when policy is parameterized by neural networks. Generally, we consider a policy network to be an MLP with well represented state features (e.g., features extracted by CNN for pixels or by LSTM for sequences) as input.

The remaining question is how we extract the policy representation from the data sources mentioned above. As shown in Figure 4, we use permutation-invariant (PI) transformations followed by an MLP to encode the data of policy $\pi$ into an embedding $\chi_\pi$ for both SPR and OPR. For SPR, each

state-action pair of $\{(s_i, a_i)\}_{i=1}^k$ is fed into a common MLP, followed by a Mean-Reduce operation on the outputted features across $k$. For OPR, we perform PI transformation (similar as done for state-action pairs) inner-layer weights and biases $\{(w_i, b_i)\}_{i=1}^h$ for each layer first, where $h$ denotes the number of nodes in this layer and $w_i, b_i$ is the income weight vector from previous layer and the bias of $i$th node; then we concatenate encoding of layers and obtain the OPR. A illustrative description for the encoding of OPR is in Figure 12 of Appendix.

To train the policy embedding $\chi_\pi$ obtained above, the most straightforward way is to backpropagate the value approximation loss of PeVFA in an *End-to-End (E2E)* fashion as illustrated on the lower-right of Figure 4. In addition, we provide two self-supervised training losses for both OPR and SPR, as illustrated on the upper-right of Figure 4. The first one is an *auxiliary loss (AUX) of policy recovery* [14], i.e., to recover the action distributions of $\pi$ from $\chi_\pi$ under different states. To be specific, an auxiliary policy decoder $\bar{\pi}(\cdot|s, \chi_\pi)$ is trained through behavioral cloning, formally to minimize cross-entropy objective $\mathcal{L}_{\text{AUX}} = -\mathbb{E}_{(s,a)}\left[\log \bar{\pi}(a|s, \chi_\pi)\right]$. For the second one, we propose to train $\chi_\pi$ by *Contrastive Learning (CL)* [54, 51]: policies are encouraged to be close to similar ones (i.e., positive samples $\pi^+$), and to be apart from different ones (i.e., negative samples $\pi^-$) in representation space. For each policy, we construct positive samples by data augmentation on policy data, depending on SPR or OPR considered; and different policies along the policy improvement path naturally provide negative samples for each other. Finally, the embedding $\chi_\pi$ is optimized through minimizing the InfoNCE loss [41] below: $\mathcal{L}_{\text{CL}} = -\mathbb{E}_{(\pi^+, \{\pi^-\})}\left[\log \frac{\exp(\chi_\pi^T W \chi_{\pi^+})}{\exp(\chi_\pi^T W \chi_{\pi^+}) + \sum_{\pi^-} \exp(\chi_\pi^T W \chi_{\pi^-})}\right]$.

Now, the training of policy representation model in Algorithm 1 can be performed with any combination of data sources and training losses provided above. A pseudo-code of the overall policy representation training framework and complete implementation details are provided in Appendix D.

# 5 Experiments

In this section, we conduct experimental study with focus on the following questions:

**Question 1** *Can value generalization offered by PeVFA improve a deep RL algorithm in practice?*

**Question 2** *Can our proposed framework to learn effective policy representation?*

Our experiments are conducted in several OpenAI Gym continuous control tasks (one from Box2D and five from MuJoCo) [6, 58]. All experimental details and curves can be found in Appendix B.

**Algorithm Implementation.** We use PPO [50] as the basic algorithm and propose a representative implementation of Algorithm 1, called **PPO-PeVFA**. PPO is a policy optimization algorithm that follows the paradigm of GPI (Figure 1, left). A value network $V_\phi(s)$ with parameters $\phi$ (i.e., conventional VFA) is trained to approximate the value of current policy $\pi$; while $\pi$ is optimized with respect to a surrogate objective [48] using advantages calculated by $V_\phi$ and GAE [49]. Compared with original PPO, PPO-PeVFA makes use of a PeVFA network $\mathbb{V}_\theta(s, \chi_\pi)$ with parameters $\theta$ rather than the conventional VFA $V_\phi(s)$, and follows the training scheme as in Algorithm 1. Note PPO-PeVFA uses the same policy optimization method as original PPO and only differs at value approximation.

**Baselines and Variants.** Except for original PPO as a default baseline, we use another two baselines: 1) PPO-PeVFA with randomly generated policy representation for each policy, denoted by **Ran PR**; 2) PPO-PeVFA with Raw Policy Representation (**RPR**), i.e., use the vector of all parameters of policy network as representation as adopted in PVFs [10]. Our variants of PPO-PeVFA differ at the policy representation used. In total, we consider 6 variants denoted by the combination of the policy data choice (i.e., **OPR**, **SPR**) and representation principle choice (i.e., **E2E**, **CL**, **AUX**).

**Experimental Details.** For all baselines and variants, we use a normal-scale policy network with 2 layers and 64 units for each layer, resulting in over 3k to 10k (e.g., Ant-v1) policy parameters depending on the environments. We do not assume the access to pre-collected policies. Thus the size of policy set increases from 1 (i.e., the initial policy) during the learning process, to about 1k to 2 for a single trial. The dimensionality of all kinds of policy representation expect for RPR is set to 64. The buffer $D$ maintains recent 200k steps of interaction experience and the policy data of corresponding policy. The number of interaction step of each trial is 1M for InvDouPend-v1 and LunarLander-v2, 4M for Ant-v1 and 2M for the others.

**Results.** The overall experimental results are summarized in Table 1. In Figure 5, we provide aggregated results across all environments expect for InvDouPend-v1 and LunarLander-v2 (since

Table 1: Average returns ($\pm$ half a std) over 10 trials for algorithms. Each result is the maximum evaluation along the training process. Top two values for each environment are bold.

| Environments | Benchmarks | | | Origin Policy Representation (Ours) | | | Surface Policy Representation (Ours) | | |
|---|---|---|---|---|---|---|---|---|---|
| | PPO | Ran PR | RPR | E2E | CL | AUX | E2E | CL | AUX |
| HalfCheetah-v1 | 2621 | 2470 | $2325 \pm 399.27$ | $3171 \pm 427.63$ | $\mathbf{3725 \pm 348.55}$ | $3175 \pm 517.52$ | $2774 \pm 233.39$ | $\mathbf{3349 \pm 341.42}$ | $3216 \pm 506.39$ |
| Hopper-v1 | 1639 | 1226 | $1097 \pm 213.47$ | $2085 \pm 310.91$ | $2351 \pm 231.11$ | $2214 \pm 360.78$ | $2227 \pm 297.35$ | $\mathbf{2392 \pm 263.93}$ | $\mathbf{2577 \pm 217.73}$ |
| Walker2d-v1 | 1505 | 1269 | $317 \pm 152.68$ | $1856 \pm 305.51$ | $2038 \pm 315.51$ | $\mathbf{2044 \pm 316.32}$ | $1930.57 \pm 456.02$ | $\mathbf{2203 \pm 381.95}$ | $1980 \pm 325.54$ |
| Ant-v1 | 2835 | 2742 | $2143 \pm 406.64$ | $3581 \pm 185.43$ | $\mathbf{4019 \pm 162.47}$ | $\mathbf{3784 \pm 268.99}$ | $3173 \pm 184.75$ | $3632 \pm 134.27$ | $3397 \pm 200.03$ |
| InvDouPend-v1 | 9344 | 9355 | $8856 \pm 551.90$ | $\mathbf{9357 \pm 0.29}$ | $9355 \pm 0.64$ | $9355 \pm 0.68$ | $9355 \pm 0.89$ | $\mathbf{9356 \pm 0.96}$ | $9355 \pm 1.42$ |
| LunarLander-v2 | 219 | 226 | $-22 \pm 35.08$ | $\mathbf{238 \pm 3.37}$ | $\mathbf{239 \pm 3.70}$ | $234 \pm 3.47$ | $236 \pm 3.13$ | $234 \pm 3.13$ | $235 \pm 5.70$ |

most algorithms achieve near-optimal results), where all returns are normalized by the results of PPO in Table 1. Full learning curves are omitted and can be found in Appendix F.2.

**To Question 1.** From Table 1, we can find that both PPO-PeVFA w/ OPR (E2E) and PPO-PeVFA w/ SPR (E2E) outperforms PPO in all 6 tasks, and achieve over 20% improvement in Figure 5. This demonstrates the effectiveness of PeVFA. Moreover, the improvement is further enlarged (to about 40%) by CL and AUX for both OPR and SPR. This indicates that the superiority of PeVFA can be further utilized with better policy representation that offers a more suitable space for value generalization.

**To Question 2.** In Table 1, consistent degeneration is observed for PPO-PeVFA w/ Ran PR due to the negative effects on generalization caused by the randomness and disorder of policy representation. This phenomenon seems to be more severe for PPO-PeVFA w/ RPR due to the complexity of high-dimensional parameter space. In contrast, the improvement achieved by our proposed PPO-PeVFA variants shows that effective policy representation can be learned from policy parameters (OPR) and state-action pairs (SPR) though value approximation loss (i.e., E2E) and further improved when additional self-supervised representation learning is involved as CL and AUX. Overall, OPR slightly outperforms SPR as CL does over AUX. We hypothesize that it is due to the stochasticity of state-action pairs which serve as inputs of SPR and training samples for AUX. This reveals the space for future improvement. In addition, we visualize the learned representation in Figure 6. We can observe that policies from different trials are locally continuous and show different modes of embedding trajectories due to random initialization and optimization; while a global evolvement among trials emerges with respect to policy performance.

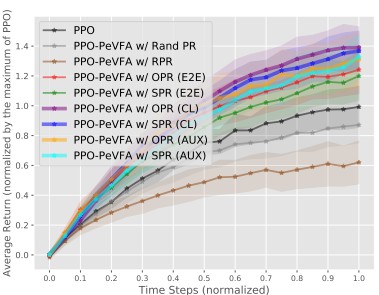

Figure 5: Normalized averaged returns aggregated over 4 MuJoCo tasks.

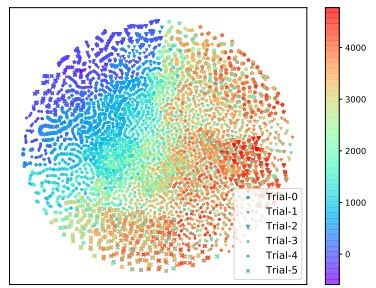

Figure 6: A t-SNE visualization for representations learned by PPO-PeVFA OPR (E2E) in Ant-v1. In total, 6k policies from 5 trials (denoted by different markers) are plotted, which are colored according to average return.

# 6 Conclusion and Future Work

In this paper, we propose Policy-extended Value Function Approximator (PeVFA) and study value generalization among policies. We propose a new form of GPI based on PeVFA which is potentially preferable to conventional VFA for value approximation. Moreover, we propose a general framework to learn low-dimensional embedding of RL policy. Our experiments demonstrate the effectiveness of the generalization characteristic of PeVFA and our proposed policy representation learning methods.

Our work opens up some research directions on value generalization among policies and policy representation. A possible future study on the theory of value generalization among policies is to consider the interplay between approximation error, policy improvement and local generalization during GPI with PeVFA. Besides, analysis on influence factors of value generalization among policies (e.g., policy representation, architecture of PeVFA) and other utilization of PeVFA are expected. For better policy representation, inspirations on OPR may be got from studies on Manifold Hypothesis of neural network; the selection of more informative state-action pairs for SPR is also worth research.

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
