# Checklist

1. For all authors...

   (a) Do the main claims made in the abstract and introduction accurately reflect the paper's contributions and scope? [Yes]

   (b) Did you describe the limitations of your work? [Yes] See the future work in Sec. 6.

   (c) Did you discuss any potential negative societal impacts of your work? [No] Our work is on general Reinforcement Learning study. No specific practical application is considered.

   (d) Have you read the ethics review guidelines and ensured that your paper conforms to them? [Yes]

2. If you are including theoretical results...

   (a) Did you state the full set of assumptions of all theoretical results? [Yes]

   (b) Did you include complete proofs of all theoretical results? [Yes]

3. If you ran experiments...

   (a) Did you include the code, data, and instructions needed to reproduce the main experimental results (either in the supplemental material or as a URL)? [No] Our experimental environment are public and standard. All the information needed to reproduce our results is provided in the main body and appendix. Code will be available publicly soon.

   (b) Did you specify all the training details (e.g., data splits, hyperparameters, how they were chosen)? [Yes] Partially in main body and all details can be found in the appendix document.

   (c) Did you report error bars (e.g., with respect to the random seed after running experiments multiple times)? [Yes]

   (d) Did you include the total amount of compute and the type of resources used (e.g., type of GPUs, internal cluster, or cloud provider)? [Yes]

4. If you are using existing assets (e.g., code, data, models) or curating/releasing new assets...

   (a) If your work uses existing assets, did you cite the creators? [Yes]

   (b) Did you mention the license of the assets? [Yes] We use a free education licence for students for MuJoCo.

   (c) Did you include any new assets either in the supplemental material or as a URL? [No]

   (d) Did you discuss whether and how consent was obtained from people whose data you're using/curating? [N/A]

   (e) Did you discuss whether the data you are using/curating contains personally identifiable information or offensive content? [N/A]

5. If you used crowdsourcing or conducted research with human subjects...

   (a) Did you include the full text of instructions given to participants and screenshots, if applicable? [N/A]

   (b) Did you describe any potential participant risks, with links to Institutional Review Board (IRB) approvals, if applicable? [N/A]

   (c) Did you include the estimated hourly wage paid to participants and the total amount spent on participant compensation? [N/A]

 **Appendix**

 # A  Supplementary Materials for Theoretical Analysis

 ## A.1  More on Definition 1 and 2

 In Definition 1, we use $\mathscr{P}_\pi$ for a formal description of value approximation process, i.e., the learning process
 of a parametrized PeVFA $\mathbb{V}_\theta$ with parameters $\theta \in \Theta$ to approximate the values of policy $\pi$. For a usual
 example, one can consider $\mathscr{P}_\pi$ as multiple times of parameter update via gradient descent with respect to
 $f_\theta(\pi) = \|\mathbb{V}_\theta(\pi) - V^\pi\|$. Note that $f_\theta(\pi)$ can be equivalent to a common value approximation loss function
 $L(\theta) = \mathbb{E}_{s\sim\rho(s)}\left(\mathbb{V}_\theta(s,\pi) - \hat{V}^\pi(s)\right)^2$ with some unbiased estimates $\hat{V}^\pi$ from experiences stored, when the
 same state distribution $\rho(s)$ is considered. Thus, with sufficient capacity of function approximation and certain
 number of training, we can expect a contraction of approximation loss for policy $\pi$ obtained by $\mathscr{P}_\pi$.

 We use Definition 2 to characterize the local smoothness of approximation loss $f_\theta$ near policy $\pi$ with Lipschitz
 continuity. Consider a typical PeVFA $V_\theta$ parameterized by an MLP with finite weights, biases and non-linear
 activation. Such a $V_\theta$ is Lipschitz continuity with a bounded Lipschitz constant, as it is made up of function
 transformations that individually have bounded Lipschitz constants, e.g., weight matrix $w$ of some layer has
 bounded Lipschitz constant to be the operator norm of matrix $w$ and ReLU activation has Lipschitz constant of
 1. Further, easily we have for any $\pi$ and $\pi'$,

$$|f_\theta(\pi) - f_\theta(\pi')| \leq \|\mathbb{V}_\theta(\pi) - \mathbb{V}_\theta(\pi')\| + \|V^\pi - V^{\pi'}\|. \tag{2}$$

 As mentioned above, $V_\theta$ is Lipschitz continuity with a bounded Lipschitz constant; and the norm of true value
 vector of two policies is also finite. Thus, $f_\theta$ in this case can also have a bounded Lipschitz constant $L$.

 ## A.2  Proof of Lemma 1

 *Proof.*    For the clarity, we also use $f$ and $\hat{f}$ as abbreviations of $f_\theta$ and $f_{\hat{\theta}}$ in the following. Start from the
 $\hat{L}$-continuity of $\hat{f}(\theta)$ (recall Definition 2), we have the upper bound of $\hat{f}(\pi_2)$ below:

$$\hat{f}(\pi_2) \leq \hat{f}(\pi_1) + \hat{L} \cdot d(\pi, \pi'). \tag{3}$$

 The second term in Equation 3 is decided by the two policies we considered and a Lipschitz constant $\hat{L}$. Moreover,
 the constant $\hat{L}$ (i.e., locality property) is related to the parameters $\hat{\theta}$ of PeVFA. In general, we denote the above
 term as $\mathcal{M}(\pi_1, \pi_2, \hat{L})$ called *locality margin*. The locality margin $\mathcal{M}(\pi_1, \pi_2, \hat{L})$ can have different forms that
 depends on the specific locality property, for examples:

$$\mathcal{M}(\pi_1, \pi_2, \hat{L}) = \begin{cases} \hat{L} \cdot d(\pi_1, \pi_2) & \text{①} \\ \langle \hat{f}'(\pi_1), \pi_2 - \pi_1 \rangle + \frac{1}{2}\hat{L} \cdot d(\pi_1, \pi_2)^2 & \text{②} \\ \langle \hat{f}'(\pi_1), \pi_2 - \pi_1 \rangle + \frac{1}{2}\langle \hat{f}''(\pi_1)(\pi_2 - \pi_1), \pi_2 - \pi_1 \rangle + \frac{1}{6}\hat{L} \cdot d(\pi_1, \pi_2)^3 & \text{③} \end{cases}$$

 ①, ②, ③ correspond to Lipschitz Continuous, Lipschitz Gradients and Lipschitz Hessian [39], which are
 conisdered in previous works on generalization studies [22, 63].

 Further, apply the Definition 1 and consider the case $f(\pi_1) \leq f(\pi_2)$, Equation 3 can be further transformed as
 follows:

$$\begin{aligned} \hat{f}(\pi_2) &\leq \hat{f}(\pi_1) + \mathcal{M}(\pi_1, \pi_2, \hat{L}) \\ &\leq \gamma f(\pi_1) + \mathcal{M}(\pi_1, \pi_2, \hat{L}) \\ &\leq \underbrace{\gamma f(\pi_2)}_{\text{generalized contraction}} + \underbrace{\mathcal{M}(\pi_1, \pi_2, \hat{L})}_{\text{locality margin}}, \end{aligned} \tag{4}$$

 which yields the generalization upper bound in Lemma 1. We note the first term of RHS of Equation 4 as
 generalized contraction term since it is from the contraction on $f(\pi_1)$ caused by the value approximation
 operator $\mathscr{P}_{\pi_1}$, and the second term as locality margin since it is determined by specific local property. $\square$

 **Remark 1** *Since value approximation is only performed for $\pi_1$, the condition $f_\theta(\pi_1) \leq f_\theta(\pi_2)$ can usually*
 *exist after a certain number of training; in turn, the complementary case $f_\theta(\pi_1) > f_\theta(\pi_2)$ is acceptable since*
 *the unoptimized approximation loss is already lower than the optimized one.*

## A.3 Proof of Corollary 1

*Proof.* Following Lemma 1, consider Lipschitz continuity for a concrete locality property of $f_{\hat{\theta}}$, we have,

$$\hat{f}(\pi_2) \leq \gamma f(\pi_2) + \hat{L} \cdot d(\pi_1, \pi_2). \tag{5}$$

Then we get the contraction condition of value generalization on $\pi_2$ in Corollary 1, by letting the RHS of Equation 5 be smaller than $f(\pi_2)$:

$$
\begin{aligned}
\gamma f(\pi_2) + \hat{L} \cdot d(\pi_1, \pi_2) &< f(\pi_2) \\
(1 - \gamma) f(\pi_2) &> \hat{L} \cdot d(\pi_1, \pi_2) \\
f(\pi_2) &> \frac{\hat{L} \cdot d(\pi_1, \pi_2)}{1 - \gamma} \geq 0.
\end{aligned}
\tag{6}
$$

$\square$

**Remark 2** *From the generalization contraction condition provided in Corollary 1, we can find that: as i. $\gamma \to 0$, or ii. $d(\pi_1, \pi_2) \to 0$, or iii. $\hat{L} \to 0$, the contraction condition is easier to achieve (or the contraction gets tighter), i.e., the generalization on unlearned policy $\pi_2$ is better.*

In another word, the tighter the contraction on learned policy $\pi_1$ is, the closer the two policies are, the smoother the approximation loss function $\hat{f}$ is, the generalization on unlearned policy $\pi_2$ is better.

Corollary 1 provides the generalization contraction condition on $f(\pi_2)$, under the assumptions that $\mathscr{P}_{\pi_1}$ is $\gamma$-contraction and $f(\pi_1) < f(\pi_2)$ (as in Lemma 1). In below, we discuss a more general condition for generalization contraction on $f(\pi_2)$ which indicates more possible cases:

**Corollary 3** *For $\theta \xrightarrow{\mathscr{P}_{\pi_1}} \hat{\theta}$ and $f_{\hat{\theta}}$ is $\hat{L}$-continuous at $\pi_1$, when $f(\pi_2) - \gamma f(\pi_1) > \hat{L} \cdot d(\pi_1, \pi_2)$, we have that $\mathscr{P}_{\pi_1}$ is also a $\gamma_g$-contraction for $\pi_2$, i.e., $f_{\hat{\theta}}(\pi_2) \leq \gamma_g f_{\theta}(\pi_2)$ with $\gamma_g \in [0, 1)$.*

*Proof.* From Equation 4, we have $\hat{f}(\pi_2) \leq \gamma f(\pi_1) + \hat{L} \cdot d(\pi_1, \pi_2)$. To yield the generalization contraction on $f(\pi_2)$, is to let

$$\hat{f}(\pi_2) \leq \gamma f(\pi_1) + \hat{L} \cdot d(\pi_1, \pi_2) < f(\pi_2), \tag{7}$$

that is to let,

$$f(\pi_2) - \gamma f(\pi_1) > \hat{L} \cdot d(\pi_1, \pi_2). \tag{8}$$

$\square$

Since $d(\pi_1, \pi_2)$ is constant in the two-policy case considered, the condition in Corollary 3 is associated to the value approximation losses on $\pi_1$ and $\pi_2$ before applying the value approximation operator $\mathscr{P}_\pi$, as well as the $\hat{L}$-continuity of $\hat{\theta}$ after applying $\mathscr{P}_\pi$. We can find similar conclusions as mentioned in Remark 2. However, Corollary 3 indicates some more cases that the condition of generalization contraction can be satisfied. For example, it can happen in the complementary cases as we assumed in Lemma 1, i.e., 1) when $f(\pi_1) > f(\pi_2)$, or 2) $\mathscr{P}_\pi$ is not a $\gamma$-contraction on $f(\pi_1)$.

## A.4 Proof of Lemma 2

*Proof.* Consider any $t \geq 0$ and $\theta_{t-1} \xrightarrow{\mathscr{P}_{\pi_t}} \theta_t$, due to $f_{\theta_t}$ is $L_t$-continuous at $\pi_t$, we have,

$$f_{\theta_t}(\pi_{t+1}) \leq f_{\theta_t}(\pi_t) + L_t \cdot d(\pi_t, \pi_{t+1}), \tag{9}$$

then due to the definition of the value approximation process $\mathscr{P}_{\pi_t}$,

$$
\begin{aligned}
f_{\theta_t}(\pi_{t+1}) &\leq \gamma_t f_{\theta_{t-1}}(\pi_t) + L_t \cdot d(\pi_t, \pi_{t+1}), \\
&= \gamma_t f_{\theta_{t-1}}(\pi_t) + \mathcal{M}_t,
\end{aligned}
\tag{10}
$$

where $\mathcal{M}_t = L_t \cdot d(\pi_t, \pi_{t+1})$. $\square$

Intuitively, such a recursive relation between the generalized approximation loss of two consecutive steps, i.e., $f_{\theta_{t-1}}(\pi_t)$ and $f_{\theta_t}(\pi_{t+1})$, are chained by the assumed continuity of the loss function $f_{\theta_t}$ and the definition of value approximation process.

### A.5 Proof of Corollary 2

*Proof.* Consider the consecutive value approximation process $\theta_{-1} \xrightarrow{\mathscr{P}_{\pi_0}} \theta_0 \xrightarrow{\mathscr{P}_{\pi_1}} \dots \xrightarrow{\mathscr{P}_{\pi_{t-1}}} \theta_{t-1} \xrightarrow{\mathscr{P}_{\pi_t}}$
$\theta_t \xrightarrow{\mathscr{P}_{\pi_{t+1}}} \dots$, following the recursive relation in Lemma 2, we have the inequality below by induction,

$$
\begin{aligned}
f_{\theta_t}(\pi_{t+1}) &\leq \gamma_t f_{\theta_{t-1}}(\pi_t) + \mathcal{M}_t, \\
&\leq \dots \\
&\leq \gamma_t \left( \gamma_{t-1} \left( \dots \left( \gamma_0 f_{\theta_{-1}}(\pi_0) + \mathcal{M}_0 \right) \dots \right) \mathcal{M}_{t-1} \right) + \mathcal{M}_t, \\
&= \underbrace{\left( \prod_{i=0}^{t} \gamma_t \right) f_{\theta_{-1}}(\pi_0)}_{\text{❶}} + \underbrace{\sum_{i=0}^{t-1} \left( \prod_{j=i+1}^{t} \gamma_j \right) \mathcal{M}_i + \mathcal{M}_t}_{\text{❷}}.
\end{aligned}
\tag{11}
$$

where $\mathcal{M}_t = L_t \cdot d(\pi_t, \pi_{t+1})$. We use ❶ to denote the term for accumulated generalized contraction of initial approximation loss and use ❷ to denote the term for accumulated locality margin. $\qquad \square$

Towards the infinity case i.e., $t \to \infty$, if we assume that *(i)* $\max_t d(\pi_t, \pi_{t+1}) < \infty$ and *(ii)* $\prod_{k=h_1}^{h_2} \gamma_k = O(\frac{1}{(h_2-h_1+1)^{1+\varepsilon}})$, $\forall 0 < h_1 \leq h_2$ with some $\varepsilon > 0$, then $\lim_{t\to\infty} f_{\theta_t}(\pi_{t+1}) < \infty$. That is because the sequence $\{\mathcal{M}_i\}_{i=0}^{t}$ has a public upper bound $\mathcal{M}_{\max} = L_{\max} \cdot \max_t d(\pi_t, \pi_{t+1})$ where $L_{\max}$ denotes the upper bound of Lipschitz constant (recall the discussion in Appendix A.1), and by (ii) $\sum_{i=0}^{t-1} \left( \prod_{j=i+1}^{t} \gamma_j \right) = O(\sum_{i=0}^{t-1} \frac{1}{(t-i+1)^{1+\varepsilon}}) < \infty$.

Note that we consider a really loose bound in the infinity case above with $\mathcal{M}_{\max}$, therefore the condition *(ii)* may be unnecessarily strict when the dynamics of $L_t$ and $d(\pi_t, \pi_{t+1})$ are considered. Intuitively, the evolvement of $L_t$ during learning process is related to function family and optimization method of $\theta_t$; and for $d(\pi_t, \pi_{t+1})$, this is related to value approximation error ($f_{\theta_t(\pi_t)}$) and policy improvement method (i.e., how $\pi_t$ is improved to be $\pi_{t+1}$). We leave these further analysis for future work.

### A.6 Proof of Theorem 1

*Proof.* By the condition in Theorem 1, we have

$$
\begin{aligned}
f_{\theta_t}(\pi_t) + f_{\theta_t}(\pi_{t+1}) &\leq \|V^{\pi_t} - V^{\pi_{t+1}}\| \\
&\leq \|\mathbb{V}_{\theta_t}(\pi_t) - V^{\pi_t}\| + \|\mathbb{V}_{\theta_t}(\pi_t) - V^{\pi_{t+1}}\| = f_{\theta_t}(\pi_t) + \|\mathbb{V}_{\theta_t}(\pi_t) - V^{\pi_{t+1}}\|,
\end{aligned}
\tag{12}
$$

where the second inequality comes from *Triangle Inequality*. Then it is straightforward that

$$
\underbrace{f_{\theta_t}(\pi_{t+1})}_{\text{generalized VAD with PeVFA}} \leq \underbrace{\|\mathbb{V}_{\theta_t}(\pi_t) - V^{\pi_{t+1}}\|}_{\text{conventional VAD}},
\tag{13}
$$

which means that with local generalization of values for successive policy $\pi_{t+1}$, the value approximation distance (VAD) can be closer in contrast to the conventional one (RHS of Equation 13). $\qquad \square$

In practice, we consider that it is also possible for farther distance to exist, e.g., the condition in above Theorem 1 is not satisfied. Moreover, under nonlinear function approximation, it is not necessary that a closer approximation distance (induced by Theorem 1) ensures easier approximation or optimization process. This can be associated to many factors, e.g., the underlying function space, the optimization landscape, the learning algorithm used and etc. In this paper, we provide a condition for potentially beneficial local generalization and we resort to empirical examination as shown in Sec. 3.3. Further investigation on the interplay between value generalization and policy learning especially under nonlinear function approximation is planned for future work.

## B   Details of Empirical Evidence of Two Kinds of Generalization

### B.1   Global Generalization in 2D Point Walker

Global generalization denotes the generalization scenario that values can generalize to unlearned policies ($\pi' \in \Pi_1$) from already learned policies ($\pi \in \Pi_0$). We conduct the following experiments to demonstrate global generalization in a 2D continuous Point Walker environment with synthetic simple policies.

**Environment.** We consider a point walker on a 2D continuous plane with:

- state: $(x, y, \sin(\theta), \cos(\theta), \cos(x), \cos(y))$, where $\theta$ is the angle of the polar coordinates,
- action: 2D displacement, $a \in \mathbb{R}^2_{[-1,1]}$,

- a deterministic transition function that describes the locomotion of the point walker, depending on the current position and displacement issued by agent, i.e., $\langle x', y' \rangle = \langle x, y \rangle + a$,

- a reward function: $r_t = \frac{u_{t+1} - u_t}{10}$ with utility $u_t = x_t^2 - y_t^2$, as illustrated in Figure 7(a).

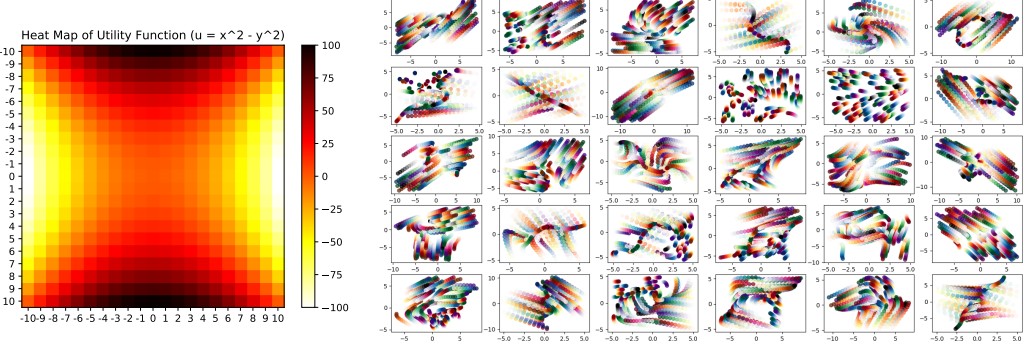

(a) Utility function heat map      (b) Examples of synthetic policy population

Figure 7: 2D Point Walker. (a) The heat map of the utility function of the 2D plane. The darker regions have higher utilities. (b) Demonstrative illustrations of trajectories generated by 30 synthetic policies, showing diverse behaviors and patterns. Each subplot illustrates the trajectories generated in 50 episodes by a randomly synthetic policy, with different colors as separation. For each trajectory (the same color in one subplot), transparency represents the dynamics along timesteps, i.e., fully transparent and non-transparent denotes the positions at first and last timesteps.

**Synthetic Policy.** We build the policy sets $\Pi = \Pi_0 \cup \Pi_1$ and $\Pi_0 \cap \Pi_1 = \emptyset$ with synthetic policies. Each synthetic policy is a 2-layer *tanh*-activated neural policy network with 2 nodes for each layer. The weights are initialized by sampling from a uniform distribution $U(-1, 1)$ and the biases are initialized by $U(-0.2, 0.2)$. Each policy is deterministic, taking an environmental state as input and outputting a displacement in the plane. We find that the synthetic population generated by such a simple way can show diverse behaviors. Figure 7(b) shows the motion patterns of an example of such a synthetic population. Note that the synthetic policies are not trained in this experiment.

**Policy Dataset.** We rollout each policy in environment to collect trajectories $\mathcal{T} = \{\tau_i\}_{i=0}^k$. For such small synthetic policies, it is convenient to obtain policy representation. Here we use the concatenation of all weights and biases of the policy network (26 in total) as representation $\chi_\pi$ for each policy $\pi$, called *raw* policy representation (RPR). Therefore, combined with the trajectories collected, we obtain the policy dataset, i.e., $\{(\chi_{\pi_j}, \mathcal{T}_{\pi_j})\}_{j=0}^n$. In total, 20k policies are synthesized in our experiments and we collected 50 trajectories with horizon 10 for each policy.

We separate the synthetic policies into training set (i.e., unknown policies $\Pi_0$) and testing set (i.e., unseen policies $\Pi_1$) in a proportion of $8 : 2$. We set a PeVFA network $\mathbb{V}_\theta(s, \chi_\pi)$ to approximate the values of training policies (i.e., $\pi \in \Pi_0$), and then conduct evaluation on testing policies (i.e., $\pi \in \Pi_1$). We use Monte Carlo return [55] of collected trajectories as approximation target (true value of policies) in this experiment. The network architecture of $\mathbb{V}_\theta(s, \chi_\pi)$ is illustrated in Figure 8(a). The learning rate is 0.005, batch size is 256. K-fold validation is performed through shuffling training and testing sets.

Figure 8(b) shows the curves of training loss and testing loss. The average losses on training and testing set are 1.782 and 2.071 over 6 trials. Figure 2(a) plots the value predictions for policies from training and testing set (100 for each). This demonstrates that a PeVFA trained with data collected by training set $\Pi_0$ achieves reasonable value prediction of unseen testing policies in $\Pi_1$. Our results indicate that value generalization can exist among policy space with a properly trained PeVFA. RPR can also be one alternative of policy representation when policy network is of small scale.

## B.2    Local Generalization in MuJoCo Continuous Control Tasks

We demonstrate local generalization of PeVFA, especially to examine the existence of Theorem 1, i.e., PeVFA can induce closer approximation distance (i.e., lower approximation error) than conventional VFA along the policy improvement path.

We use a 2-layer 8-unit policy network trained by PPO [50] algorithm in OpenAI MuJoCo continuous control tasks. As in previous section, using a very small policy network is for the convenience of training and acquisition

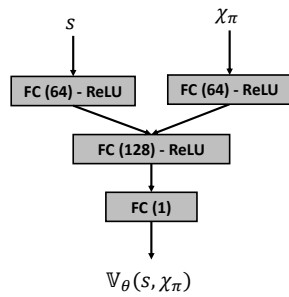
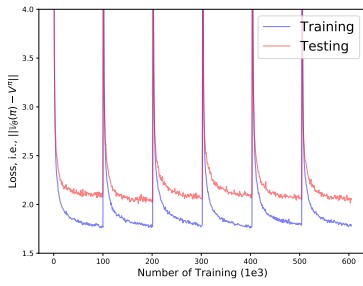

(a) Structure of PeVFA network   (b) Global Generalization on 2D Point Walker

Figure 8: Global generalization of PeVFA on 2D Point Walker. (a) An illustration of architecture of PeVFA network. FC is abbreviation for Fully-connected layer. (b) Training and testing losses. Data shuffle sand network re-initialization are performed per 100 steps, i.e., 1e5 training times.

of policy representation in this demonstrative experiment. We use all weights and biases of the small policy network (also called *raw* policy representation, RPR), whose number is about 10 to 100 in our experiments, depending on the specific environment (i.e., the state and action dimensions). We train the small policy network as commonly done with PPO [50] and GAE [49]. The conventional value network $V_\phi(s)$ (VFA), is a 2-layer 128-unit ReLU-activated MLP with state as input and value as output. Parallel to the conventional VFA in PPO, we set a PeVFA network $\mathbb{V}_\theta(s, \chi_\pi)$ with RPR as additional input. The structure of PeVFA differs at the first hidden layer which has two input streams and each of them has 64 units, as illustrated in Figure 8(a), so that making VFA and PeVFA have similar scales of parameter number. In contrast to conventional VFA $V_\phi$ which approximates the value of current policy (e.g., Algorithm 2), PeVFA $\mathbb{V}_\theta(s, \chi_\pi)$ has the capacity to preserve values of multiple policies and thus is additionally trained to approximate the values of all historical policies ($\{\pi_i\}_{i=0}^t$) along the policy improvement path (e.g., Algorithm 3). The learning rate of policy is 0.0001 and the learning rate of value function approximators ($V_\phi(s)$ and $\mathbb{V}_\theta(s, \chi_\pi)$) is 0.001. The training scheme of PPO policy here is the same as that described in Appendix F.1 and Table 2.

Note that $\mathbb{V}_\theta(\chi_\pi)$ does not interfere with PPO training here, and is only referred as a comparison with $V_\phi$ on the approximation error to the true values of successive policy $\pi_{t+1}$. We use the MC returns of on-policy data (i.e., trajectories) collected by current successive policy as unbiased estimates of true values, similarly done in [61, 12]. Then we calculate the approximation error for VFA $V_\phi$ and PeVFA $\mathbb{V}_\theta(\chi_\pi)$ to the approximation target before and after value network training of current iteration. Finally, we compare the approximation error between VFA and PeVFA to approximately examine local generalization and closer approximation target in Theorem 1. Complete results of local generalization across all 7 MuJoCo tasks are show in Figure 9. The results show that PeVFA consistently shows lower losses (i.e., closer to approximation target) across all tasks than conventional VFA before and after policy evaluation along policy improvement path, which demonstrates Theorem 1. Moreover, we also provide similar empirical evidence when policy is updated with larger learning rates in $\{0.0001, 0.001, 0.005\}$, as in Figure 10.

A common observation across almost all results in Figure 9 and in Figure 10 is that the larger the extent of policy change (see the regions with a sheer slope on green curves), the higher the losses of conventional VFA tend to be (see the peaks of red curves), where the generalization tends to be better and more significant (see the blue curves). Since InvertedPendulum-v1 is a simple task while the complexity of the solution for Ant-v1 is higher, the difference between value approximation losses of PeVFA and VFA is more significant at the regions with fast policy improvement. Besides, the Raw Policy Representation (RPR) we used here does not necessarily induce a smooth and efficient policy representation space, among which policy values are easy to generalize and optimize. Thus, RPR may be sufficient for a good generalization in InvertedPendulum-v1 but may be not in Ant-v1. Overall, we think that the quantity of value approximation loss is related to several factors of the environment such as the reward scale, the extent of policy change, the complexity of underlying solution (e.g., value function space) and some others. A further investigation on this can be interesting.

## C   Generalized Policy Iteration with PeVFA

### C.1   Comparison between Conventional GPI and GPI with PeVFA

A graphical comparison of conventional GPI and GPI with PeVFA is shown in Figure 1. Here we provide another comparison with pseudo-codes.

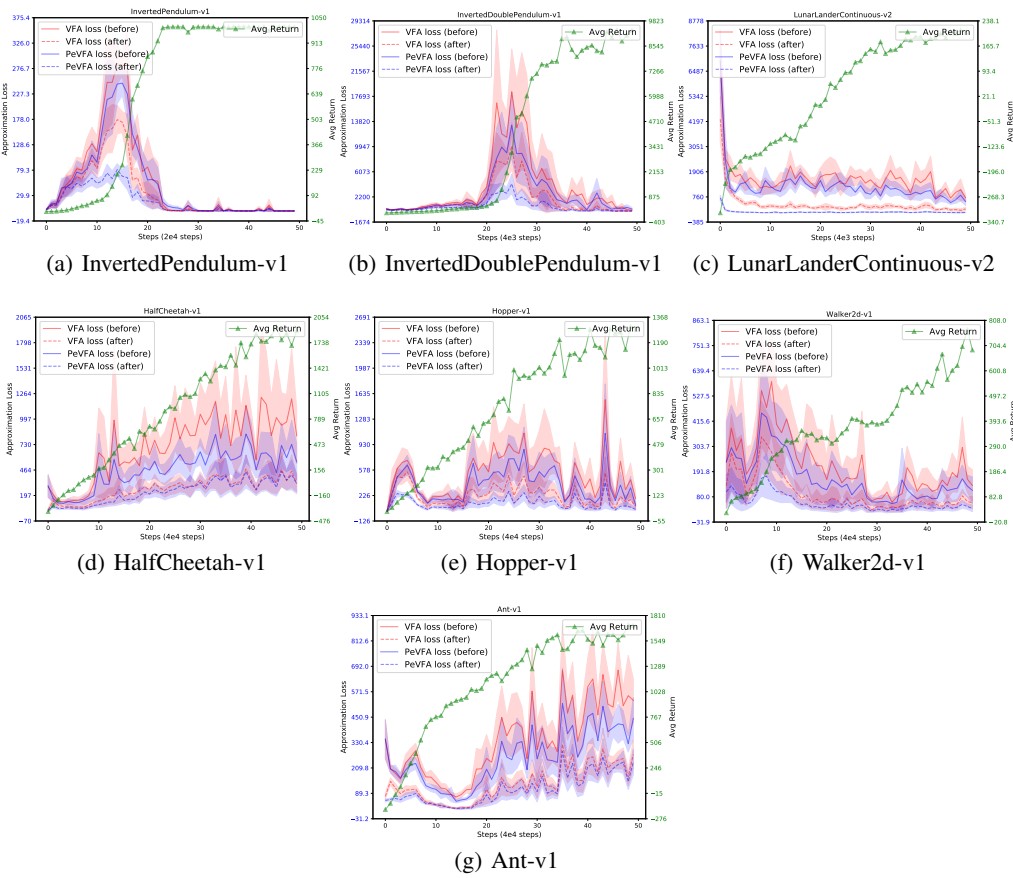

Figure 9: Complete empirical evidence of local generalization of PeVFA across 7 MuJoCo tasks. The learning rate of policy and value function approximators are 0.0001 and 0.001 respectively. Each plot has two vertical axes, the left one for approximation error (red and blue curves) and the right one for average return (green curves). Red and blue denotes the approximation error of conventional VFA ($V_\phi(s)$) and of PeVFA ($\mathbb{V}_\theta(s, \chi_\pi)$) respectively; solid and dashed curves denote the approximation error before and after the training for values of successive policy (i.e., policy evaluation) with conventional VFA and PeVFA, averaged over 6 trials. The shaded region denotes half a standard deviation of average evaluation. PeVFA consistently shows lower losses (i.e., closer to approximation target) across all tasks than conventional VFA before and after policy evaluation along policy improvement path, which demonstrates Theorem 1.

From the lens of Generalized Policy Iteration [55], for most model-free policy-based RL algorithms, the approximation of value function and the update of policy through policy gradient theorem are usually conducted iteratively. Representative examples are REINFORCE [55], Advantage Actor-Critic [36], Deterministic Policy Gradient (DPG) [52] and Proximal Policy Optimization (PPO) [50]. With conventional value function (approximator), policy evaluation is usually performed in an on-policy or off-policy fashion. We provide a general GPI description of model-free policy-based RL algorithm with conventional value functions in Algorithm 2.

Note that we use subscript $t - 1 \to t$ (Line 13 in Algorithm 2) to let the updated value functions to correspond to the evaluated policy $\pi_t$ during policy evaluation process in current iteration.

As a comparison, a new form of GPI with PeVFA is shown in Algorithm 3. Except for the different parameterization of value function, PeVFA can perform additionally training on historical policy experiences at each iteration (Line 7-8). This is naturally compatible with PeVFA since it develops the capacity of conventional value function to preserve the values of multiple policies. Such a training is to improve the value generalization of PeVFA among a policy set or policy space. Note that for value approximation of current policy $\pi_t$ (Line 10-14), the start points are generalized values of $\pi_t$ from historical approximation, i.e., $\mathbb{V}_{t-1}(s, \chi_{\pi_t})$ and $\mathbb{Q}_{t-1}(s, a, \chi_{\pi_t})$. In another word, this is the place where local generalization steps (illustrated in Figure 2(b)) are. One may compare with conventional start points ($V_{t-1}^\pi(s)$ and $Q_{t-1}^\pi(s, a)$, Line 13 in Algorithm 2) and see the difference,

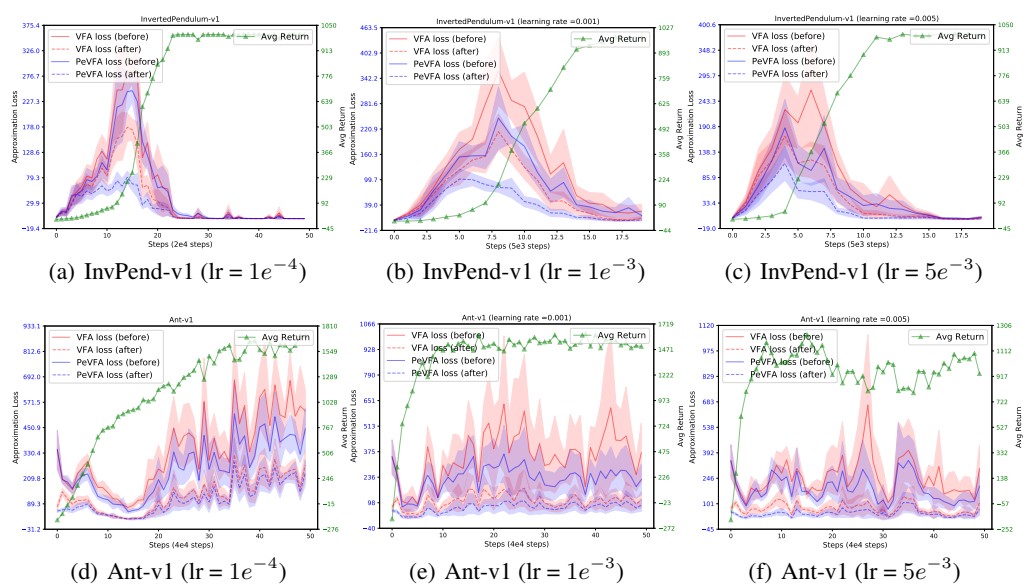

Figure 10: Empirical evidence of local generalization of PeVFA on InvertedPendulum-v1 and Ant-v1 with different learning rates of policy, i.e., $\{0.0001, 0.001, 0.005\}$. Results are averaged over 6 trials.

---

**Algorithm 2** Generalized policy iteration for model-free policy-based RL algorithm with conventional value functions ($V^\pi(s)$ or $Q^\pi(s, a)$)

1: Initialize policy $\pi_0$ and $V_{-1}^\pi(s)$ or $Q_{-1}^\pi(s, a)$
2: Initialize experience buffer $\mathcal{D}$
3: **for** iteration t $= 0, 1, 2, \ldots$ **do**
4:     Rollout policy $\pi_t$ in the environment and obtain trajectories $\mathcal{T}_t = \{\tau_i\}_{i=0}^k$
5:     Add experiences $\mathcal{T}_t$ in buffer $\mathcal{D}$
6:     **if** *on-policy update* **then**
7:         Prepare training samples from rollout trajectories $\mathcal{T}_t$
8:     **else if** *off-policy update* **then**
9:         Prepare training samples by sampling from buffer $\mathcal{D}$
10:     **end if**
11:     Calculate approximation target $\{y_i\}_i$ from training samples (e.g., with MC or TD)
12:     # Generalized Policy Evaluation
13:     Update $V_{t-1}^\pi(s)$ or $Q_{t-1}^\pi(s, a)$ with $\{(s_i, y_i)\}_i$ or $\{(s_i, a_i, y_i)\}_i$, i.e., $V_t^\pi \longleftarrow V_{t-1}^\pi$ or $Q_t^\pi \longleftarrow Q_{t-1}^\pi$
14:     # Generalized Policy Improvement
15:     Update policy $\pi_t$ with regard to $V_t^\pi(s)$ or $Q_t^\pi(s, a)$ through some policy gradient theorem, i.e., $\pi_{t+1} \longleftarrow \pi_t$
16: **end for**

---

e.g., $V_{t-1}^\pi(s) \Leftrightarrow V^{\pi_{t-1}}(s) \Leftrightarrow \mathbb{V}_{t-1}(s, \chi_{\pi_{t-1}})$ is different with $\mathbb{V}_{t-1}(s, \chi_{\pi_t})$, where $\Leftrightarrow$ is used to denote an equivalence in definition. As discussed in Sec. 3.3 and 3.4, we suggest that such local generalization steps help to reduce approximation error and thus improve efficiency during the learning process.

## C.2    More Discussions on GPI with PeVFA

**Off-Policy Learning.** Off-policy Value Estimation [55] denotes to evaluate the values of some target policy from data collected by some behave policy. As commonly seen in RL (also shown in Line 6-10 in Algorithm 2), different algorithms adopt on-policy or off-policy methods. For GPI with PeVFA, especially for the value estimation of historical policies (Line 8 in Algorithm 3), on-policy and off-policy methods can also be considered here. One interesting thing is, in off-policy case, one can use experiences from any policy for the learning of another one, which can be appealing since the high data efficiency of value estimation of each policy can

**Algorithm 3** Generalized policy iteration of model-free policy-based RL algorithm with PeVFAs ($\mathbb{V}(s, \chi_\pi)$ or $\mathbb{Q}(s, a, \chi_\pi)$)

---

1: Initialize policy $\pi_0$ and PeVFA $\mathbb{V}_{-1}(s, \chi_\pi)$ or $\mathbb{Q}_{-1}(s, a, \chi_\pi)$
2: Initialize experience buffer $\mathcal{D}$
3: **for** iteration t $= 0, 1, 2, \ldots$ **do**
4:      Rollout policy $\pi_t$ in the environment and obtain trajectories $\mathcal{T}_t = \{\tau_i\}_{i=0}^k$
5:      Get the policy representation $\chi_{\pi_t}$ for policy $\pi_t$ (from policy network parameters or policy rollout experiences)
6:      Add experiences $(\chi_{\pi_t}, \mathcal{T}_t)$ in buffer $\mathcal{D}$
7:      # Value approximation training for historical policies $\{\pi_i\}_{i=0}^{t-1}$
8:      Update PeVFA $\mathbb{V}_{t-1}(s, \chi_{\pi_i})$ or $\mathbb{Q}_{t-1}(s, a, \chi_{\pi_i})$ with all historical policy experiences $\{(\chi_{\pi_i}, \mathcal{T}_i)\}_{i=0}^{t-1}$
9:      # Conventional value approximation training for current policy $\pi_t$
10:      **if** *on-policy update* **then**
11:          Update PeVFA $\mathbb{V}_{t-1}(s, \chi_{\pi_t})$ or $\mathbb{Q}_{t-1}(s, a, \chi_{\pi_t})$ for $\pi_t$ with on-policy experiences $(\chi_{\pi_t}, \mathcal{T}_t)$
12:      **else if** *off-policy update* **then**
13:          Update PeVFA $\mathbb{V}_{t-1}(s, \chi_{\pi_t})$ or $\mathbb{Q}_{t-1}(s, a, \chi_{\pi_t})$ for $\pi_t$ with off-policy experiences $\chi_{\pi_t}$ and $\{\mathcal{T}_i\}_{i=0}^t$ from experience buffer $\mathcal{D}$
14:      **end if**
15:      $\mathbb{V}_t \longleftarrow \mathbb{V}_{t-1}$ or $\mathbb{Q}_t \longleftarrow \mathbb{Q}_{t-1}$
16:      Update policy $\pi_t$ with regard to $\mathbb{V}_t(s, \chi_{\pi_t})$ or $\mathbb{Q}_t(s, a, \chi_{\pi_t})$ through some policy gradient theorem, i.e., $\pi_{t+1} \longleftarrow \pi_t$
17: **end for**

---

strengthen value generalization among themselves with PeVFA, which further improve the value estimation process.

**Convergence of GPI with PeVFA.** Convergence of GPI is usually discussed in some ideal cases, e.g., with small and finite state action spaces and with sufficient function approximation ability. In this paper, we focus on the comparison between conventional VFA and PeVFA in value estimation, i.e., Policy Evaluation, and we make no assumption on the Policy Improvement part. We conjecture that with the same policy improvement algorithm and sufficient function approximation ability, GPI with conventional VFA and GPI with PeVFA finally converge to the same policy. Moreover, based on Theorem 1 and our empirical evidence in Sec. 3.3, GPI with PeVFA can be more efficient in some cases: with local generalization, it could take less experiences (training) for PeVFA to reach the same level of approximation error than conventional VFA, or with the same amount of experience (training), PeVFA could achieve lower approximation error than conventional VFA. We believe that a deeper dive in convergence analysis is worth further investigation.

**PeVFA with TD Value Estimation.** In this paper, we propose PPO-PeVFA as a representative instance of re-implementing DRL algorithms with PeVFA. Our theoretical results and algorithm 3 proposed under the general policy iteration (GPI) paradigm are suitable for TD value estimation as well in principle. One potential thing that deserves further investigation is that, it can be a more complex generalization problem since the approximation target of TD learning is moving (in contrast to the stationary target when unbiased Monte Carlo estimates are used). The non-stationarity induced by TD is recognized to hamper the generalization performance in RL as pointed out in recent work [21]. Further study on PeVFA with TD learning (e.g., TD3 and SAC) is planned in the future as mentioned in Sec. 6.

# D   Policy Representation Learning Details

## D.1   Policy Geometry

A policy $\pi \in \Pi = \mathcal{P}(\mathcal{A})^\mathcal{S}$, defines the behavior (action distribution) of the agent under each state. For a more intuitive view, we consider the geometrical shape of a policy: all state $s \in \mathcal{S}$ and all action $a \in \mathcal{A}$ are arranged along the $x$-axis and $y$-axis of a 2-dimensional plane, and the probability (density) $\pi(a|s)$ is the value of $z$-axis over the 2-dimensional plane. Note that for finite state space and finite action space (discrete action space), the policy can be viewed as a $|\mathcal{S}| \times |\mathcal{A}|$ table with each entry in it is the probability of the corresponding state-action case. Without loss of generality, we consider the continuous state and action space and the policy geometry here. Illustrations of policy geometry examples are shown in Figure 11.

778 Figure 11(a) shows the policy geometry in a general case, where the policy can be defined arbitrarily. Generally,
779 the policy geometry can be any possible geometrical shape (s.t. $\forall s \in \mathcal{S}, \sum_{a \in \mathcal{A}} \pi(a|s) = 1$). This means that
780 the policy geometry is not necessarily continuous or differentiable in a general case. Specially, one can imagine
781 that the geometry of a deterministic policy consists of peak points ($z = 1$) for each state and other flat regions
782 ($z = 0$). Figure 11(b) shows an example of synthetic continuous policy which can be viewed as a 3D curved
783 surface. In Deep RL, a policy may usually be modeled as a deep neural network. Assume that the neural policy
784 is a function that is almost continuous and differentiable everywhere, the geometry of such a neural policy can
785 also be continuous and differentiable almost everywhere. As shown in Figure 11(c), we provide a demo of neural
786 policy by smoothing an arbitrary policy along both state and action axes.

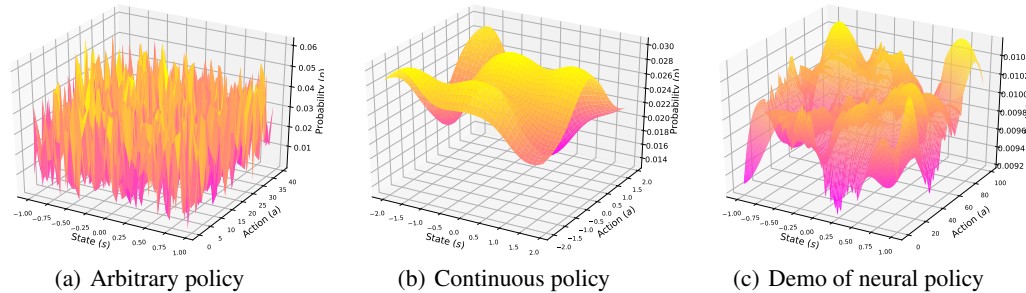

(a) Arbitrary policy          (b) Continuous policy         (c) Demo of neural policy

Figure 11: Examples of policy geometry. (a) An arbitrary policy, where $p(s, a)$ is sampled from $\mathcal{N}(0, 1)$ for a joint space of 40 states and 40 actions and then normalized along action axis. States are squeezed into the range of $[-1, 1]$ for clarity. (b) A synthetic continuous policy with $p(s, a) = (1 - a^5 + s^5) \exp(-s^2 - a^2)$ for a joint space of $s \in [-2, 2]$ and $a \in [-2, 2]$ (each of which are discretized into 40 ones) and then normalized along action axis. (c) A general demo of neural network policy, generated from an arbitrary policy (as in (a)) over a joint space of 200 states and 100 actions with some smoothing skill. States are squeezed into the range of $[-1, 1]$ for clarity and the probability masses of actions under each state are normalized to sum into 1.

### D.2 Implementation Details of Surface Policy Representation (SPR) and Origin Policy Representation (OPR)

789 Here we provide a detailed description of how to encode different policy data for Surface Policy Representation
790 (SPR) and Origin Policy Representation (OPR) we introduced in Sec. 4.

791 **Encoding of State-action Pairs for SPR.** Given a set of state-action pairs $\{s_i, a_i\}_{i=1}^n$ (with size $[n, s\_dim +$
792 $a\_dim]$) generated by policy $\pi$ (i.e., $a_i \sim \pi(\cdot|s_i)$), we concatenate each state-action pair and obtain an
793 embedding of it by feeding it into an MLP, resulting in a stack of state-action embedding with size $[n, e\_dim]$.
794 After this, we perform a mean-reduce operator on the stack and obtain an SPR with size $[1, e\_dim]$. A similar
795 permutation-invariant transformation is previously adopted to encode trajectories in [14].

796 **Encoding of Network Parameters for OPR.** We propose a novel way to learn low-dimensional embedding
797 from policy network parameters directly. To our knowledge, we are the first to learn policy embedding from
798 neural network parameters in RL. Note that latent space of neural networks are also studied in differentiable
799 Network Architecture Search (NAS) [32, 33], where architecture-level embedding are usually considered. In
800 contrast, OPR cares about parameter-level embedding with a given architecture.

801 Consider a policy network to be an MLP with well-represented state (e.g., CNN for pixels, LSTM for sequences)
802 as input and deterministic or stochastic policy output. We compress all the weights and biases of the MLP to
803 obtain an OPR that represents the decision function. The encoding process of an MLP with two hidden layers
804 is illustrated in Figure 12. The main idea is to perform permutation-invariant transformation for inner-layer
805 weights and biases for each layer first. For each unit of some layer, we view the unit as a non-linear function of
806 all outputs, determined by weights, a bias term and activation function. Thus, the whole layer can be viewed
807 as a batch of operations of previous outputs, e.g., with the shape $[h_t, h_{t-1} + 1]$ for $t \geq 1$ and $t = 0$ is also for
808 the input layer. Note that we neglect activation function in the encoding since we consider the policy network
809 structure is given. That is also why we call OPR as parameter-level embedding in contrast to architecture-level
810 enbedding in NAS (mentioned in the last paragraph). We then feed the operation batch in an MLP and perform
811 mean-reduce to outputs. Finally we concatenate encoding of layers and obtain the OPR.

812 We use permutation-invariant transformation for OPR because that we suggest the operation batch introduced
813 in the last paragraph can be permutation-invariant. Actually, our encoding shown in Figure 12 is not strict to
814 obtain permutation-invariant representation since inter-layer dependencies are not included during the encoding

process. We also tried to incorporate the order information during OPR encoding and we found similar results with the way we present in Figure 12, which we adopt in our experiments.

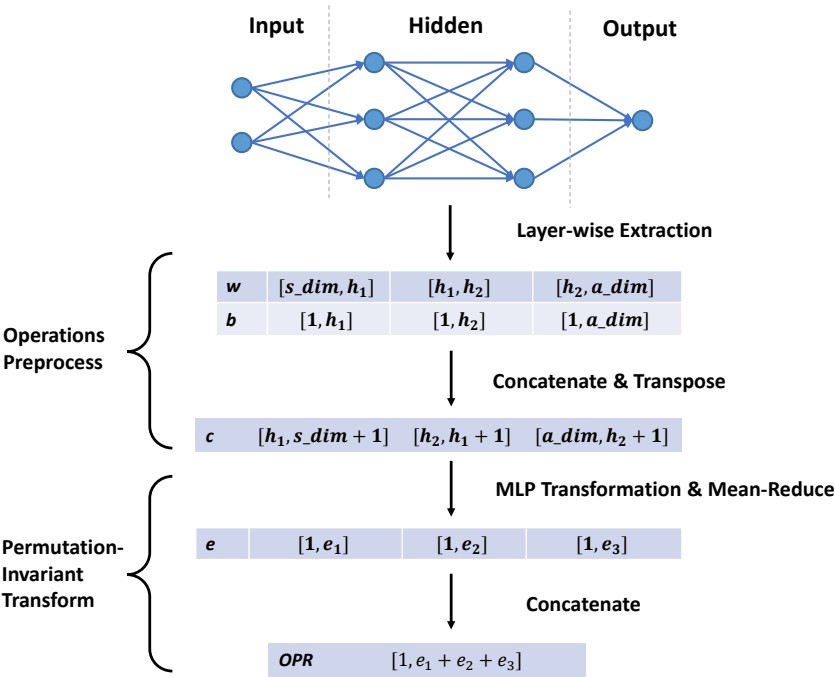

Figure 12: An illustration for policy encoder of Origin Policy Representation (OPR) for a two-layer MLP. $h_1, h_2$ denotes the numbers of hidden units for the first and second hidden layers respectively. The main idea is to perform permutation-invariant transformation for inner-layer weights and biases for each layer first and then concatenate encoding of layers.

**Towards more sophisticated RL policy that operates images.** Our proposed two policy representations (i.e., OPR and SPR) can basically be applied to encode policies that operate images, with the support of advanced image-based state representation. For OPR, a policy network with image input usually has a pixel feature extractor like Convolutional Neural Networks (CNNs) followed by a decision model (e.g., an MLP). With effective features extracted, the decision model can be of moderate (or relatively small) scale. Recent works on unsupervised representation learning like MoCo [20], SimCLR [7], CURL [54] also show that a linear classifier or a simple MLP which takes compact representation of images learned in an unsupervised fashion is capable of solving image classification and image-based continuous control tasks. In another direction, it is promising to develop more efficient even gradient-free OPR, for example using the statistics of network parameters in some way instead of all parameters as similarly considered in [60].

For SPR, to encode state-action pairs (or sequences) with image states can be converted to the encoding in the latent space. The construction of latent space usually involves self-supervised representation learning, e.g., image reconstruction, dynamics prediction. A similar scenario can be found in recent model-based RL like Dreamer [16], where the imagination is efficiently carried out in the latent state space rather than among original image observations.

Overall, we believe that there remain more effective approaches to represent RL policy to be developed in the future in a general direction of OPR and SPR, which are expected to induce better value generalization in a different RL problems.

## D.3 Data Augmentation for SPR and OPR in Contrastive Learning

Data augmentation is studied to be an important component in contrastive learning in deep RL recently [24, 28]. Contrastive learning usually resorts to data augmentation to build positive samples. Data augmentation is typically performed on pixel inputs (e.g., images) problems [20, 7]. In our work, we train policy representation with contrastive learning where data augmentation is performed on policy data. For SPR, i.e., state-action pairs as policy data, there is no need to perform data augmentation since different batches of randomly sampled state-action pairs naturally forms positive samples, since they all reflect the behavior of the same policy. A similar idea can also be found in [11] when dealing with task context in Meta-RL.

For OPR, i.e., policy network parameters as policy data, it is unclear how to perform data augmentation on them. In this work, we consider two kinds of data augmentation for policy network parameters as shown in Figure 13. We found similar results for both random mask and noise corruption, and we use random mask as default data augmentation in our experiments.

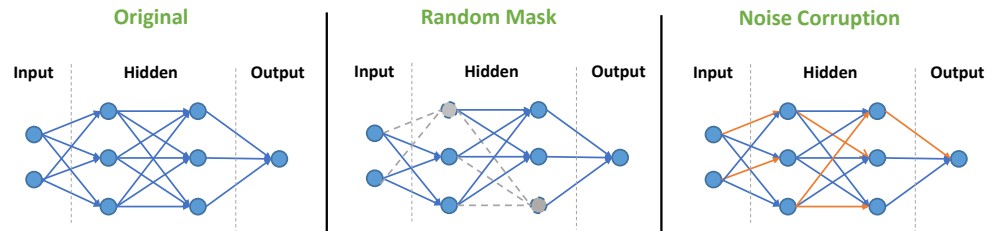

Figure 13: Examples of data augmentation on policy network parameters for Origin Policy Representation (OPR). *Left*: an example of original policy network. *Middle*: dropout-like random masks are performed on original policy network, where gray dashed lines represent the weights masked out. *Right*: randomly selected weights are corrupted by random noises, denoted by orange lines.

As an unsupervised representation learning method, contrastive Learning encourages policies to be close to similar ones (i.e., positive samples $\pi^+$) and to be apart from different ones (i.e., negative samples $\pi^-$) in policy representation space. The policy representation network is then trained with InfoNCE loss [41], i.e., to minimize the cross-entropy loss below:

$$\mathcal{L}_{\text{CL}} = -\mathbb{E}\left[\log \frac{\exp(\chi_\pi^T W \chi_{\pi^+})}{\exp(\chi_\pi^T W \chi_{\pi^+}) + \sum_{\pi^-} \exp(\chi_\pi^T W \chi_{\pi^-})}\right]$$

## D.4 Pseudo-code of Policy Representation Learning Framework

The pseudo-code of the overall framework of policy representation learning is in Algorithm 4. The policy representation learning is conducted base on a policy dataset, which stores the policy data, i.e., interaction trajectories generated by policies and the parameters of policy networks. Such a dataset can be obtained in different ways, e.g., pre-collected, online collected and etc. In our experiments, we do not assume the access to pre-collected or given policy data; instead, we use the data of all historical policies met along the policy improvement path during the online learning process.

Different kinds of policy data (i.e., state-action pairs or policy parameters) are used depending on the policy representation adopted (i.e., SPR or OPR). For policy representation learning, the value function approximation loss (E2E) is used as a default choice of training loss in our framework. In addition, the auxiliary loss (AUX) of policy recovery and contrastive learning (CL) serve as another two options to be optimized for representation learning. Note that in Line 21, the positive samples $\chi_{\pi_i^+}$ is obtained from a momentum policy encoder [20] with another augmentation for corresponding policy data, while negative samples $\chi_{\pi_i^-}$ are other policy embeddings in the same batch, i.e., $\chi_{\pi_i^-} \in B \backslash \{\chi_{\pi_i}\}$.

## D.5 Criteria of A Good Policy Representation

To answer the question: what is a good representation for RL policy ought to be? We assume the following criteria:

- *Dynamics*. Intuitively, a good policy representation should contain the information of how the policy influences the environment (dynamics and rewards).

- *Consistency*. A good policy representation should keep the consistency among both policy space and presentation space. Concretely, the policy representation should be distinguishable, i.e., different policies also differ among their representation. In contrast, the representation of similar polices should lay on the close place in the representation space.

- *Geometry*. Additionally, from the lens of policy geometry as shown in Appendix D.1, a good policy representation should be an reflection of policy geometry. It should show a connection to the policy geometry or be interpretable from the geometric view.

From the perspective of above criteria, SPR follows *Dynamics* and *Geometry* while OPR may render them in an implicit way since network parameters determine the nonlinear function of policy. Auxiliary loss for policy

**Algorithm 4** A Framework of Policy Representation Learning

**Input:** policy dataset $\mathbb{D} = \{(\pi_i, \omega_i, \mathcal{D}_{\pi_i})\}_{i=1}^n$, consisting of policy $\pi_i$, policy parameters $\omega_i$ and state-action pairs $\mathcal{D}_{\pi_i} = \{(s_j, a_j)\}_{j=1}^m$

1: Initialize the policy encoder $g_\alpha$ with parameters $\alpha$
2: Initialize the policy decoder (or master policy) (network) $\bar{\pi}_\beta(a|s, \chi_\pi)$ for SPR and the weight matrix $W$ for ORP respectively
3: **for** iteration $i = 0, 1, 2, \ldots$ **do**
4:    Sample a mini-batch of policy data $\mathcal{B}$ from $\mathbb{D}$
5:    # Encode and obtain the policy embedding $\chi_{\pi_i}$ with SPR or OPR
6:    **if** Use OPR **then**
7:        **if** Use Contrastive Learning **then**
8:            Perform data augmentation on each $w_i \in \mathcal{B}$
9:        **end if**
10:        $\chi_{\pi_i} = g_\alpha^{\text{OPR}}(\omega_{\pi_i})$ for each $(\pi_i, \omega_i, \cdot) \in \mathcal{B}$
11:    **else if** Use SPR **then**
12:        $\chi_{\pi_i} = g_\alpha^{\text{SPR}}(B_i)$ where $B_i$ is a mini-batch of state-action pairs sampled from $\mathcal{D}_{\pi_i}$, for each $(\pi_i, \cdot, \mathcal{D}_{\pi_i}) \in \mathcal{B}$
13:    **end if**
14:    # Train policy encoder $g_\alpha$ in different ways (i.e., AUX or CL)
15:    **if** Use Auxiliary Loss (AUX) **then**
16:        Sample a mini-batch of state-action pairs $B = (s_i, a_i)_{i=1}^b$ from $\mathcal{D}_{\pi_i}$ for each $\pi_i$
17:        Compute the auxiliary loss, $\mathcal{L}_{\text{Aux}} = -\sum_{(s_i, a_i) \in B} \log \bar{\pi}_\alpha(a_i|s_i, \chi_{\pi_i})$
18:        Update parameters $\alpha, \beta$ to minimize $\mathcal{L}^{\text{Aux}}$
19:    **end if**
20:    **if** Use Contrastive Learning (CL) **then**
21:        Calculate contrastive loss, $\mathcal{L}_{\text{CL}} = -\sum_{\chi_{\pi_i} \in B} \log \frac{\exp(\chi_{\pi_i}^T W \chi_{\pi_i^+})}{\exp(\chi_{\pi_i}^T W \chi_{\pi_i^+}) + \sum_{\pi_i^-} \exp(\chi_{\pi_i}^T W \chi_{\pi_i^-})}$, where $\chi_{\pi_i^+}, \chi_{\pi_i^-}$ are positive and negative samples
22:        Update parameters $\alpha, W$ to minimize $\mathcal{L}^{\text{CL}}$
23:    **end if**
24:    # Train policy encoder $g_\alpha$ with the PeVFA approximation loss (E2E)
25:    Calculate the value approximation loss of PeVFA, $\mathcal{L}_{\text{Val}}$
26:    Update parameters $\alpha$ to minimize $\mathcal{L}_{\text{Val}}$
27: **end for**

recovery (AUX) is a learning objective to acquire *Dynamics*; Contrastive Learning (CL) is used to impose *Consistency*.

Based on the above thoughts, we hypothesize the reasons of several findings as shown in the comparison in Table 1. First, AUX naturally overlaps with SPR and OPR to some degree for *Dynamics* while CL is relatively complementary to SPR and OPR for *Consistency*. This may be the reason why CL improves the E2E representation more than AUX in an overall view. Second, the noise of state-action samples for SPR may be the reason to OPR's slightly better overall performance than that of SPR (similar results are also found in our visualizations as in Figure 19).

Moreover, the above criteria are mainly considered from an unsupervised or self-supervised perspective. However, a sufficiently good representation of all the above properties may not be necessary for a specific downstream generalization or learning problem which utilizes the policy representation. A problem-specific learning signal, e.g., the value approximation loss in our paper (E2E representation), can be efficient since it is to extract the most relevant information in policy representation for the problem. A recent work [59] also studies the relation between self-supervised representation and downstream tasks from the lens of mutual information. Therefore, we suggest that a trade-off between good unsupervised properties and efficient problem-specific information of policy representation should be considered when using policy representation in a specific problem.

## E  Complete Background and Detailed Related Work

### E.1  Reinforcement Learning

**Markov Decision Process.** We consider a Markov Decision Process (MDP) defined as $\langle \mathcal{S}, \mathcal{A}, r, \mathcal{P}, \gamma, \rho_0 \rangle$ with $\mathcal{S}$ the state space, $\mathcal{A}$ the action space, $r$ the reward function, $\mathcal{P}$ the transition function, $\gamma \in [0, 1)$ the discount factor and $\rho_0$ the initial state distribution. A policy $\pi \in \Pi = P(\mathcal{A})^{|\mathcal{S}|}$, defines the distribution over all actions for each state. The agent interacts with the environment with its policy, generating the trajectory $s_0, a_0, r_1, s_1, a_1, r_2, ..., s_t, a_t, r_{t+1}, ...$, where $r_{t+1} = r(s_t, a_t)$. An RL agent seeks for an optimal policy that maximizes the expected long-term discounted return, $J(\pi) = \mathbb{E}_{s_0 \sim \rho_0, a \sim \pi} \left[ \sum_{t=0}^{\infty} \gamma^t r_{t+1} \right]$.

**Value Function.** Almost all RL algorithms involve value functions [55], which estimate how good a state or a state-action pair is conditioned on a given policy. The *state-value function* $v^\pi(s)$ is defined in terms of the expected return obtained through following the policy $\pi$ from a state $s$:

$$v^\pi(s) = \mathbb{E}_\pi \left[ \sum_{t=0}^{\infty} \gamma^t r_{t+1} | s_0 = s \right] \text{ for all } s \in \mathcal{S}.$$

Similarly, *action-value function* is defined for all state-action pairs as $q^\pi(s, a) = \mathbb{E}_\pi \left[ \sum_{t=0}^{\infty} \gamma^t r_{t+1} | s_0 = s, a_0 = a \right]$. Typically, value functions are learned through Monte Carlo (MC) or Temporal Difference (TD) algorithms [55].

Bellman equations defines the recursive relationships among value functions. The *Bellman Expectation equation* of $v^\pi(s)$ has a matrix form as below [55]:

$$V^\pi = r^\pi + \gamma \mathcal{P}^\pi V^\pi = (\mathcal{I} - \gamma \mathcal{P}^\pi)^{-1} r^\pi, \tag{14}$$

where $V^\pi$ is a $|\mathcal{S}|$-dimensional vector, $\mathcal{P}^\pi$ is the state-to-state transition matrix $\mathcal{P}^\pi(s'|s) = \sum_{a \in \mathcal{A}} \pi(a|s)\mathcal{P}(s'|s, a)$ and $r^\pi$ is the vector of expected rewards $r^\pi(s) = \sum_{a \in \mathcal{A}} \pi(a|s)r(s, a)$. Equation 14 indicates that value function is determined by policy $\pi$ and environment models (i.e., $\mathcal{P}$ and $r$. For a conventional value function, all of them are modeled implicitly within a table or a function approximator, i.e., a mapping from only states (and actions) to values.

**Generalized Policy Iteration.** Sutton and Barto [55] consider most RL algorithms can be described in the paradigm of Generalized Policy Iteration (GPI). In recent decade, RL algorithms usually resort to function approximation (e.g., deep neural networks) to deal with large and continuous state space. An illustration of GPI with function approximation is on the left of Figure 1. We use $\theta$ to denote the parameters of parameterized value functions. Without loss of generality, we do not plot the parameters of policy since it is not necessary for parameterized policy to exist, e.g., value-based RL algorithms [35]. For policy evaluation, value function approximators are updated in finite times to approximate the true values (i.e., $V_\theta(s) \to v^\pi(s)$, $Q_\theta(s, a) \to q^\pi(s, a)$), yet can never be perfect. For policy improvement, the policy are improved with respected to the approximated value functions in an implicit (e.g., value-based RL) or explicit way (policy-based RL). In deep RL, perfect policy evaluation and effective policy improvement are non-trivial to obtain with complex non-linear function approximation from deep neural networks, thus most convergence and optimality results in conventional RL usually no longer hold. From these two aspects, many works study how to improve the value function approximation [61, 4, 27] and to propose more effective policy optimization or search algorithms [48, 50, 15].

### E.2  A Unified View of Extensions of Conventional Value Function from the Vector Form of Bellman Equation

Recall the vector form of Bellman equation (Equation 14), it indicates that value function is a function of policy $\pi$ and environmental models (i.e., $\mathcal{P}$ and $r$). In conventional value functions and approximators, only state (and action) is usually taken as input while other components in Equation 14 are modeled implicitly. Beyond state (and action), consider explicit representation of some of components in Equation 14 during value estimation can develop the ability of conventional value functions in different ways, to solve challenging problems, e.g., goal-conditioned RL [45, 1], Hierarchical RL [37, 64], opponent modeling and ad-hoc team [19, 14, 57], and context-based Meta-RL [43, 29].

Most extensions of conventional VFA mentioned above are proposed for the purpose of value generalization (among different space). Therefore, we suggest such extensions are derived from the same start point (i.e., Equation 14) and differ at the objective to represent and take as additional input explicitly of conventional value functions. We provide a unified view of such extensions below:

- Goal-conditioned RL and context-based meta-RL usually focus on a series of tasks with similar goals and environment models (i.e., $\mathcal{P}$ and $r$). With goal representation as input, usually a subspace of state space [45, 1], a value function approximation (VFA) can generalize values among goal space. Similarly, with context representation [43, 11, 42], values generalize in meta tasks.

- Opponent modeling, ad-hoc team [19, 14, 57] seek to generalize among different opponents or teammates in a Multiagent System, with learned representation of opponents. This can be viewed as a special case of value generalization among environment models since from one agent view, other opponents are part of the environment which also determines the dynamics and rewards. In multiagent case, one can expand and decompose the corresponded joint policy in Equation 14 to see this.

- Hierarchical RL is also a special case of value generalization among environment models. In goal-reaching fashioned Hierarchical HRL [37, 30, 38], high-level controllers (policy) issue goals for low-level controls at an abstract temporal scale, while low-level controls take goals also as input and aim to reach the goals. For low-level policies, a VFA with a given or learned goal representation space can generalize values among different goals, similar to the goal-conditioned RL case as discussed above. Another perspective is to view the separate learning process of hierarchical policies for different levels as a multiagent learning system. Recently, a work [64] follows this view and extends high-level policy with representation of low-level learning.

The common thing of above is that, they learn a representation of the environment (we call *external variables*). In contrast, we study value generalization among agent's own policies in this paper, which cares about *internal variables*, i.e., the learning dynamics inside of the agent itself.

**Relation between PeVFA Value Approximation and Context-based Meta-RL.** For a given MDP, performing a policy in the MDP actually induces a Markov Reward Process (MRP) [55]. One can view the policy and actions are absorbed in the transition function of MRP. A value function defines the expected long-term returns starting from a state. Therefore, different policies induces different MRPs and PeVFA value approximation can be considered as a meta prediction task. In analogy to context-based Meta-RL where a task context is learned to capture the underlying transition function and reward function of a MDP (i.e., task), one can view policy representation as the context of corresponding MRP, since it is the underlying variable that determines the transition function of MRPs.

## E.3  A Review of Works on Policy Representation/Embedding Learning

Recent years, a few works involve representation or embedding learning for RL policy [18, 14, 3, 42, 64, 17]. We provide a brief review and summary for above works below.

The most common way to learn a policy representation is to extract from interaction trajectories through policy recovery (i.e., behavioral cloning). For Multiagent Opponent Modeling [14], a policy representation is learned from interaction episodes (i.e., state-action trajectories) through a *generative loss* and *discriminate loss*. Generative loss is the same as the policy recovery auxiliary loss; discriminate loss is a triplet loss that minimize the representation distance of the same policy and maximize those of different ones, which has the similar idea of Contrastive Learning [41, 54]. Such opponent policy representations are used for prediction of interaction outcomes for ad-hoc teams and are taken in policy network for some learning agent to facilitate the learning when cooperating or competing with unknown opponents. More recently, in Hierarchical RL [64], a representation is learned to model the low-level policy through *generative loss* mentioned above. The low-level policy representation is taken in high-level policy to counter the non-stationarity issue of co-learning of hierarchical policies. Later, Raileanu et al. [42] resort to almost the same method and the learned policy representation is taken in their proposed PDVF. Along with a task context, the policy for a specific task can be optimized in policy representation space, inducing a fast adaptation in new tasks. In summary, such a representation learning paradigm can be considered as Surface Policy Representation (SPR) for policy data encoding (trajectories as a special form of state-action pairs) plus policy recovery auxiliary loss (AUX) as we introduced in Sec. 4.

A recent work [17] proposes Policy Evaluation Network (PVN) to approximate objective function $J(\pi)$. We consider PVN as an predecessor of PDVF we mentioned above since offline policy optimization is also conducted in learned representation space in a single task after similarly well training the PVN on many policies. The authors propose *Network Fingerprint* to represent policy network. To circumvent the difficulty of representing the parameters directly, policy network outputs (policy distribution) under a set of *probing states* are concatenated and then taken as policy representation. Such probing states are randomly sampled for initialization and also optimized with gradients through PVN and policies, like a search in joint state space. In principle, we also consider this as a special instance of SPR, because network fingerprint follows the idea of reflecting the information of how policy can behave under some states. Intuitively from a geometric view, this can be viewed as using the concatenation of several representative (as denoted by the probing states) cross-sections in policy surface (e.g., Figure 11) to represent a policy. For another view, one can imagine an equivalent case between SPR and network fingerprint, when state-action pairs of a deterministic policy are processed in SPR and a representation consisting of a number of actions under some key states or representative states is used in network fingerprint. Two potential issues may exist for network fingerprint. First, the dimensionality of representation is proportional to the number of probing states (i.e., $n|\mathcal{A}|$), where a dilemma exists: more probing states are more representative while dimensionality can increase correspondingly. Second, it can be non-trivial and even

1005 unpractical to optimize probing states in the case with relatively state space of high dimension, which introduces
1006 additional computational complexity and optimization difficulty.

1007 In another concurrent work [10], a class of Parameter-based Value Functions (PVFs) are proposed. Instead of
1008 learning or designing a representation of policy, PVFs simply parse all the policy weights as inputs to the value
1009 function (i.e., Raw Policy Representation as also mentioned in our paper), even in the nonlinear case. Apparently,
1010 this can result in a unnecessarily large representation space which increase the difficulty of approximation and
1011 generalization. The issues of naively flattening the policy into a vector input are also pointed out in PVN [17].

1012 Others, several works in Policy Adaptation and Transfer [18, 3], Gaussian policy embedding representations are
1013 construct through Variantional Inference.

## F  Experimental Details and Complete Results

### F.1  Experimental Details

1016 **Environment.** We conduct our experiments on commonly adopted OpenAI Gym[1] continuous control tasks
1017 [6, 58]. We use the OpenAI Gym with version 0.9.1, the mujoco-py with version 0.5.4 and the MuJoCo products
1018 with version `MJPRO131`. Our codes are implemented with Python 3.6 and `Tensorflow`.

1019 **Resources and Equipment Used.** Our experiments are mainly conducted on a NVIDIA GeForce RTX 2080
1020 Ti with 11 GB memory. A single run of PPO-PeVFA usually takes 3-4 hours with about 6 trials are running
1021 simultaneously on the same GPU.

1022 **Implementation.** We use Proximal Policy Optimization (PPO) [50] with Generalized Advantage Estimator
1023 (GAE) [49] as our baseline algorithm. Recent works [8, 2] point out code-level optimizations influence the
1024 performance of PPO a lot. For a fair comparison and clear evaluation, we perform no code-level optimization
1025 in our experiments, e.g., state standardization, reward scaling, gradient clipping, parameter sharing and etc.
1026 Our proposed algorithm PPO-PeVFA is implemented based on PPO, which only differs at the replacement
1027 for conventional value function network with PeVFA network. Policy network is a 2-layer MLP with 64
1028 units per layer and ReLU activation, outputting a Gaussian policy, i.e., a tanh-activated mean along with a
1029 state-independent vector-parameterized log standard deviation. For PPO, the conventional value network $V_\phi(s)$
1030 (VFA) is a 2-layer 128-unit ReLU-activated MLP with state as input and value as output. For PPO-PeVFA, the
1031 PeVFA network $\mathbb{V}_\theta(s, \chi_\pi)$ takes as input state and policy representation $\chi_\pi$ which has the dimensionality of 64,
1032 with the structure illustrated in Figure 8(a). We do not use parameter sharing between policy and value function
1033 approximators for more clear evaluation.

1034 **Training and Evaluation.** We use Monte Carlo returns for value approximation. In contrast to conventional
1035 VFA $V_\phi$ which approximates the value of current policy (e.g., Algorithm 2), PeVFA $\mathbb{V}_\theta(s, \chi_\pi)$ is additionally
1036 trained to approximate the values of all historical policies ($\{\pi_i\}_{i=0}^t$) along the policy improvement path (e.g.,
1037 Algorithm 3). The policy network parameterized by $\omega$ is then updated with following loss function:

$$\mathcal{L}^{\text{PPO}}(\omega) = -\mathbb{E}_{\pi_{\omega^-}} \left[ \min \left( \rho_t \hat{A}(s_t, a_t), \text{clip}(\rho_t, 1 - \epsilon, 1 + \epsilon) \hat{A}(s_t, a_t) \right) \right], \tag{15}$$

1038 where $\hat{A}(s_t, a_t)$ is advantage estimation of old policy $\pi_{\omega^-}$, which is calculated by GAE based on conventional
1039 VFA $V_\phi$ or PeVFA $\mathbb{V}_\theta(s, \chi_\pi)$ respectively, and $\rho_t = \frac{\pi_\omega(a_t, s_t)}{\pi_{\omega^-}(a_t, s_t)}$ is the importance sampling ratio. Note that
1040 both PPO and PPO-PeVFA update the policy according to Equation 15 and only differ at advantage estimation
1041 based on conventional VFA $V_\phi$ or PeVFA $\mathbb{V}_\theta(s, \chi_\pi)$. This ensures that the performance difference comes only
1042 from different approximation of policy values. Common learning parameters for PPO and PPO-PeVFA are
1043 shown in Table 2. For each iteration, we update value function approximators first and then the policy with
1044 updated values. Such a training scheme is used for both PPO and PPO-PeVFA. For evaluation, we evaluate
1045 the learning algorithm every 20k time steps, averaging the returns of 10 episodes. Fewer evaluation points are
1046 selected and smoothed over neighbors and then plotted in our learning curves below.

1047 **Details for PPO-PeVFA.** For PeVFA, the training process also involves value approximation of historical
1048 policies and learning of policy representation. Training details are shown in Table 3. PeVFA $\mathbb{V}_\theta(s, \chi_\pi)$ is
1049 trained every 10 steps with a batch of 64 samples from an experience buffer with size of 200k steps. Policy
1050 representation model is trained at intervals of 10 or 20 steps depending on OPR or SPR adopted. Due to 1k - 2k
1051 policies are collected in total in each trial, a relatively small batch size of policy is used. For OPR, Random
1052 Mask (Figure 13) is performed on all weights and biases of policy network except for the output layer (i.e., mean
1053 and log-std). For SPR, two buffers of state-action pairs are maintained for each policy: a small one is sampled
1054 for calculating SPR and the relatively larger one is sampled for auxiliary training (policy recovery).

---

[1]`http://gym.openai.com/`

Table 2: Common hyperparameter choices of PPO and PPO-PeVFA.

| Hyperparameters for PPO & PPO-PeVFA | |
| --- | --- |
| Policy Learning Rate | $10^{-4}$ |
| Value Learning Rate | $10^{-3}$ |
| Clipping Range Parameter ($\epsilon$) | 0.2 |
| GAE Parameter ($\lambda$) | 0.95 |
| Discount Factor ($\gamma$) | 0.99 |
| On-policy Samples Per Iteration | 5 episodes or 2000 time steps |
| Batch Size | 128 |
| Actor Epoch | 10 |
| Critic Epoch | 10 |
| Optimizer | Adam |

Table 3: Training details for PPO-PeVFA, including value approximation of historical policies and policy representation learning. CL is abbreviation for Contrastive Learning and AUX is for auxiliary loss of policy recovery. In our experiments, grid search is performed for the best hyperparamter configuration regarding terms with multiple alternatives (i.e., {}).

| Value Approximation for Historical Policies | |
| --- | --- |
| Value Learning Rate | $10^{-3}$ |
| Training Frequency | Every 10 time steps |
| Batch Size | 64 |
| Experience Buffer Size | 200k (steps) |
| Policy Representation Learning | |
| Training Frequency | Every {10, 20} time steps |
| Policy Num Per Batch | {16, 32} |
| SPR $s, a$ Pair Num | {200, 500} |
| CL Learning Rate | {$10^{-3}, 5 \cdot 10^{-4}$} |
| CL Momentum | {$5 \cdot 10^{-2}, 10^{-2}, 5 \cdot 10^{-3}$} |
| CL Mask Ratio for OPR | {0.1, 0.2} |
| CL Sample Ratio for SPR | 0.8 |
| AUX Learning Rate | $10^{-3}$ |
| AUX Batch Size | {128, 256} |

## F.2 Complete Learning Curves for Evaluation Results in Table 1

Corresponding to Table 1, an overall view of learning curves of all variants of PPO-PeVFA as well as baseline algorithms are shown in Figure 17. One can refer to Figure 14 for a clearer view of the effects of PeVFA (with E2E policy representation), and Figure 15, 16 for the effects of self-supervised policy representations, i.e., CL and AUX.

## F.3 Visualization of Learned Policy Representation

To show how the learned representation is like in a low-dimensional space, we visualize the learned representation of policies encountered during the learning process.

**Visualization Design.** We record all policies on the policy improvement path during the learning process of a PPO-PeVFA agent. For each trial in our experiments in MuJoCo continuous control tasks, about 1k - 2k policies are collected. We run 5 trials and 5k - 12k policies are collected in total for each task. We also store the policy representation model at intervals for each trial, and we use last three checkpoints to compute the representation of each policy collected. For each policy collected during 5 trials, its representation for visualization is obtained by averaging the results of 3 checkpoints of each trial and then concatenating the results from 5 trials. Finally, we plot 2D embedding of policy representations prepared above through t-SNE [34] and Principal Component Analysis (PCA) in `sklearn`[2].

---

[2]`https://scikit-learn.org/stable/index.html`

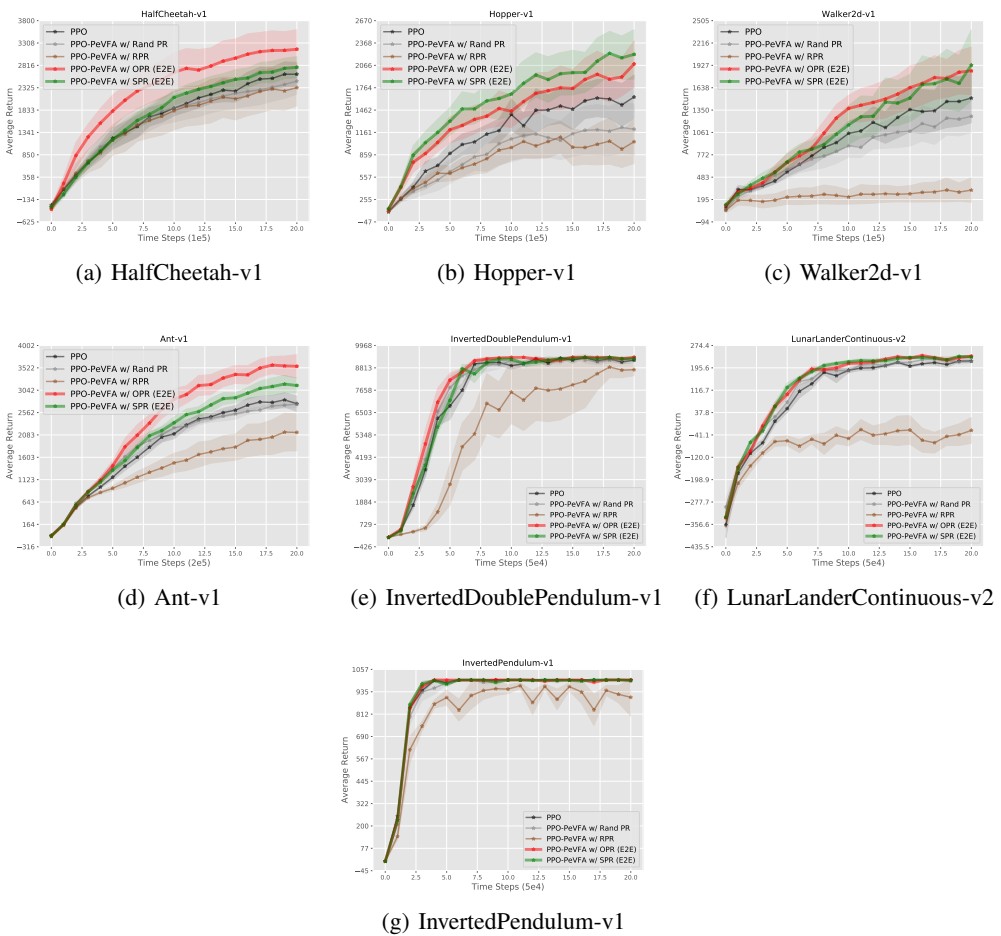

Figure 14: Evaluations of PPO-PeVFA with end-to-end (E2E) trained OPR and SPR in MuJoCo continuous control tasks. The results demonstrate the effectiveness of PeVFA and two kinds of policy representation, answering the Question 1. The results are average returns and the shaded region denotes half a standard deviation over 10 trials.

**Results and Analysis.** Visualizations of OPR and SPR learned in an end-to-end fashion in HalfCheetah-v1 and Ant-v1 are in Figure 18 and 19. We use different types of markers to distinguish policies from different trials to see how policy evolves in representation space from different random initialization. Moreover, we provide two views: performance view and process view, to see how policies are aligned in representation space regarding performance and 'age' of policies respectively.

Visualization of OPR trained in end-to-end fashion is shown in Figure 18. From the performance view, it is obvious that policies of poor and good performances are aligned from left to right in t-SNE representation space and are aligned at two distinct directions in PCA representation space. An evolvement of policies from different trials can be observed in subplot (b) and (d). Thus, policies from different trials are locally continuous; while policies are globally consistent in representation space with respect to policy performance. Moreover, we can observe multimodality for policies with comparable performance. This means that the obtained representation not only reflects optimality information but also maintains the behavioral characteristic of policy.

Parallel to OPR, end-to-end trained SPR is visualized in Figure 19. A more obvious multimodality can be observed in both t-SNE and PCA space: policies from different trials start from the same region and then diverge during the following learning process. Different from OPR, SPR shows more distinction among different trials since SPR is a more direct reflection of policy behavior (*dynamics* property as mentioned in Sec. D.5). Another thing is, policies from different trials forms wide 'strands' especially in t-SNE representation space. We conjecture that it is because SPR is a more stochastic way to obtain representation as random selected state-action pairs are used.

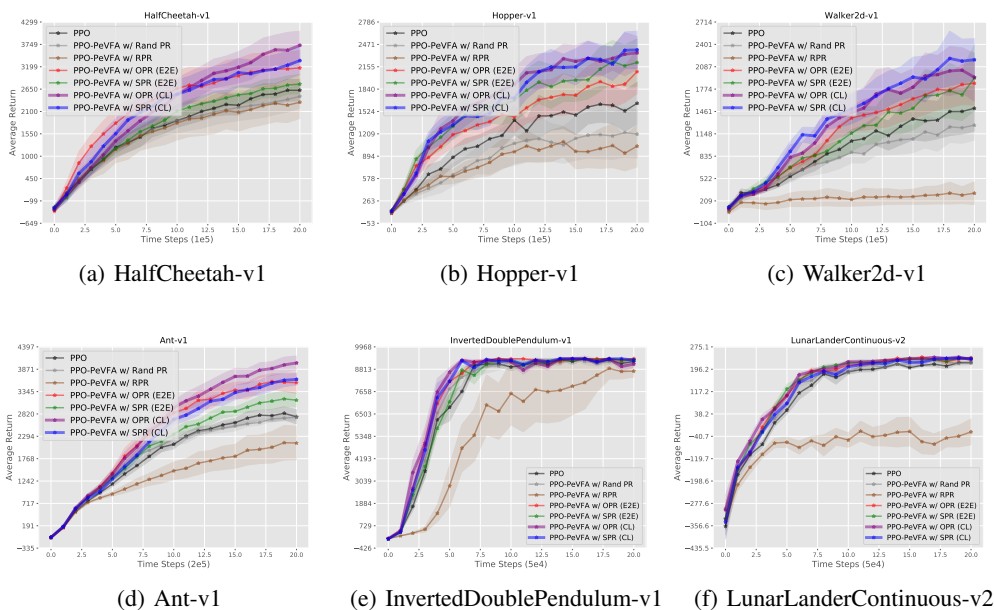

Figure 15: Evaluations of PPO-PeVFA with OPR and SPR trained through contrastive learning (CL) in MuJoCo continuous control tasks. The results are average returns and the shaded region denotes half a standard deviation over 10 trials.

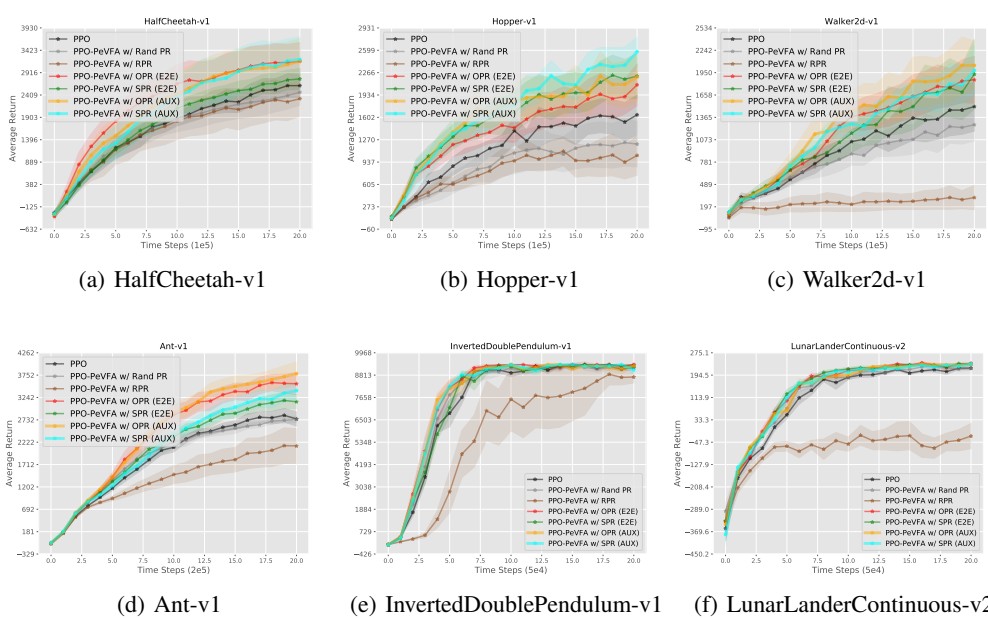

Figure 16: Evaluations of PPO-PeVFA with OPR and SPR trained through auxiliary loss of policy recovery (AUX) in MuJoCo continuous control tasks. The results are average returns and the shaded region denotes half a standard deviation over 10 trials.

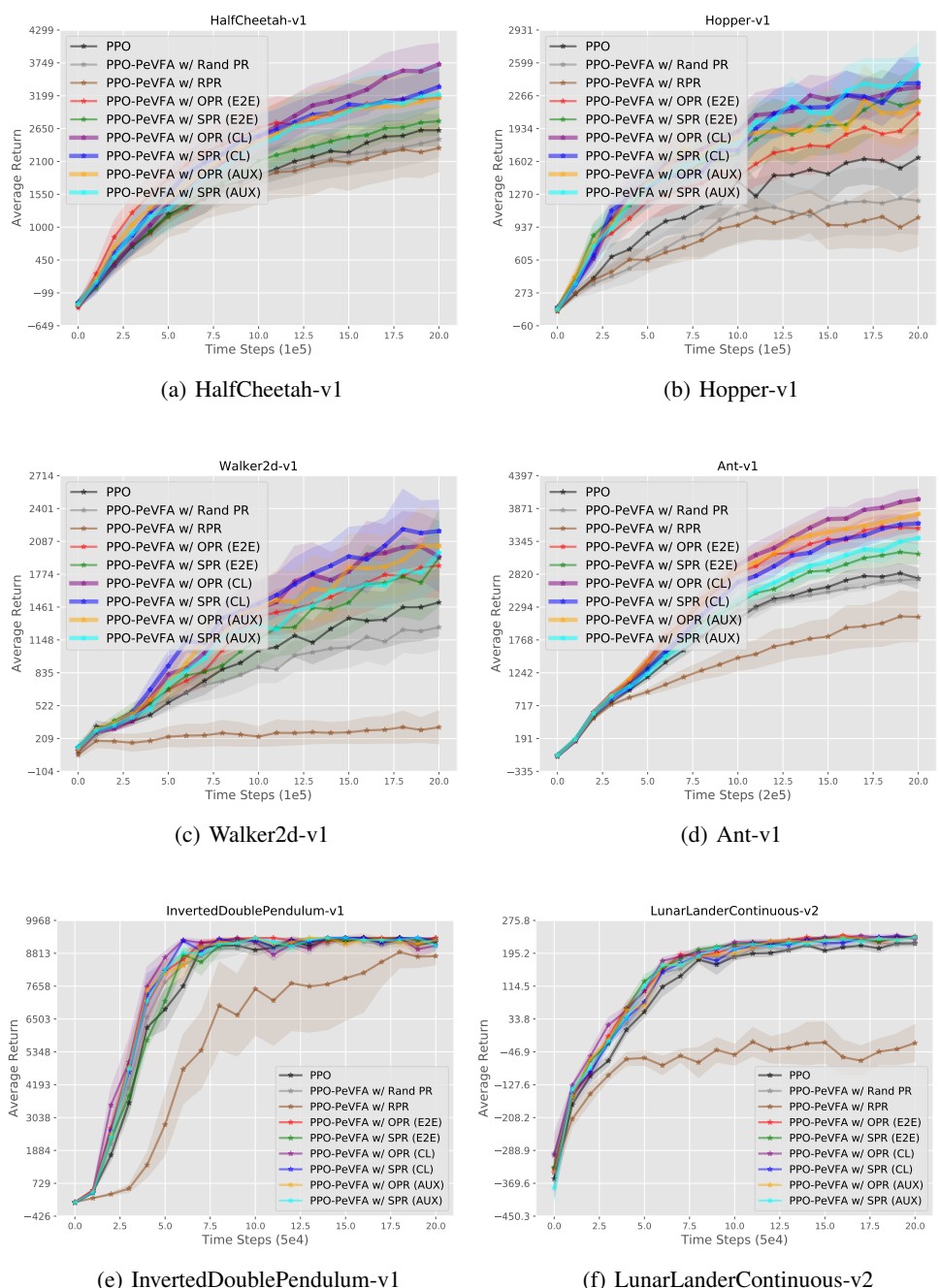

Figure 17: An overall view of performance evaluations of different algorithms in MuJoCo continuous control tasks. The results are average returns and the shaded region denotes half a standard deviation over 10 trials.

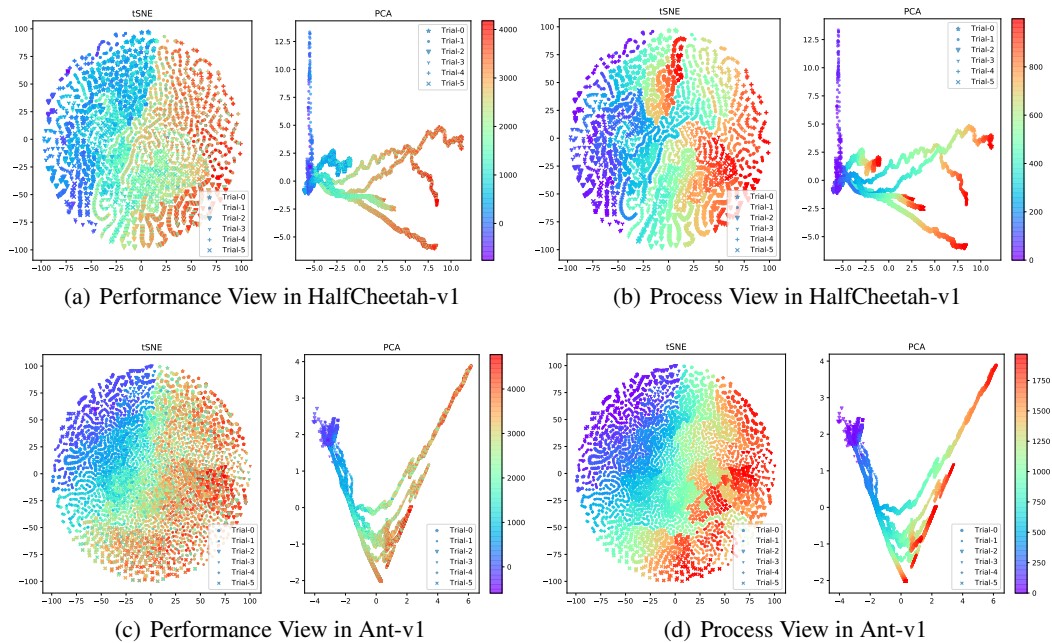

(a) Performance View in HalfCheetah-v1    (b) Process View in HalfCheetah-v1

(c) Performance View in Ant-v1    (d) Process View in Ant-v1

Figure 18: Visualizations of end-to-end (E2E) learned Origin Policy Representation (OPR) for policies collected during 5 trials (denoted by different kinds of markers). In total, about 6k policies are plotted for HalfCheetah-v1 (*a-b*) and 12k for Ant-v1 (*c-d*). In each subplot, t-SNE and PCA 2D embeddings are at left and right respectively. In performance view, each policy (i.e., marker) is colored by its performance evaluation (averaged return). In process view, each policy is colored by its corresponding iteration ID during GPI process.

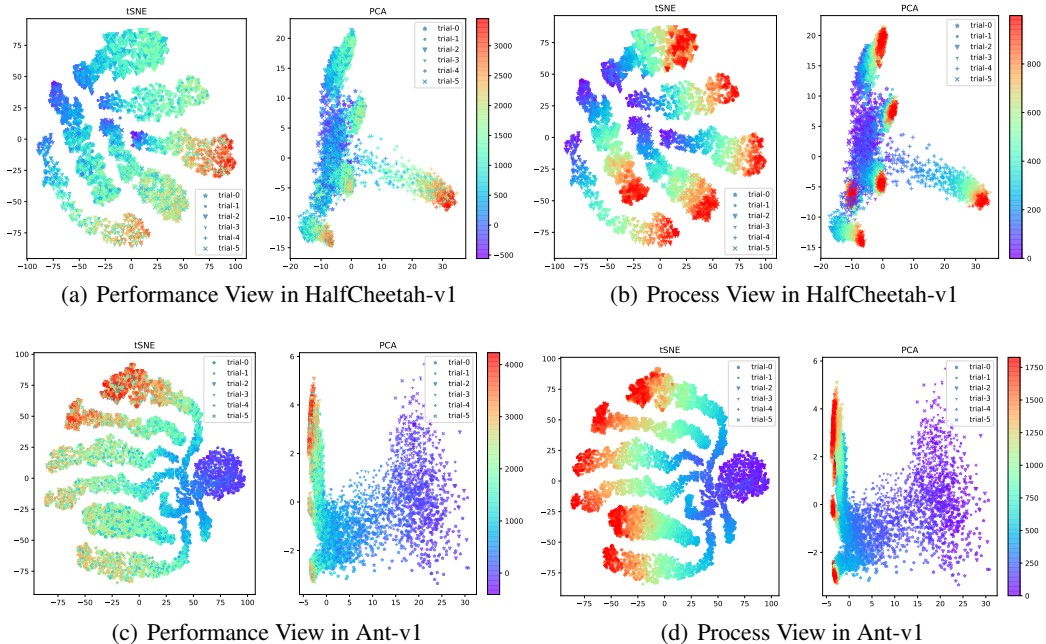

(a) Performance View in HalfCheetah-v1    (b) Process View in HalfCheetah-v1

(c) Performance View in Ant-v1    (d) Process View in Ant-v1

Figure 19: Visualizations of end-to-end (E2E) learned Surface Policy Representation (SPR) for policies collected during 5 trials (denoted by different kinds of markers). In performance view, each policy (i.e., marker) is colored by its performance evaluation (averaged return). In process view, each policy is colored by its corresponding iteration ID during GPI process.