# OpenReview forum: "Represent Your Own Policies: Reinforcement Learning with Policy-extended Value Function Approximator"
_NeurIPS.cc/2021/Conference — NeurIPS 2021 Submitted_

### Official Review · Reviewer_aYpG · 2021-07-15

**Rating:** 5
**Confidence:** 4

**Summary:**

This paper studies value function approximators that explicitly take the policy as input, which the authors named policy-extended value function approximator (PeVFA). This is different from the common value function approximators which implicitly represent the value of the current policy. The authors discuss two scenarios where PeVFA could provide potential benefits via generalization: global generalization where the policies are sampled from a fixed distribution and local generalization where the policies are sampled from the policy improvement path. The latter is a typical scenario for policy improvement. The authors provide a theoretical analysis of the generalization properties of PeVFA. To apply PeVFA in practice, the authors also investigate several different ways of learning a compact embedding for policies: end-to-end learning, auxiliary loss of policy recovery, and contrastive learning. Empirical study shows that PeVFA improves over the conventional value function approximators on the Mujoco continuous control benchmark. The empirical results also provide some insight into the effectiveness of different policy embedding learning methods.

**Limitations And Societal Impact:**

This work is not directly related to any real-world applications thus is far from making any negative societal impact at the current stage.

**Main Review:**

Things to like:
* 1. Overall the paper is written in a coherent manner. The motivation is clear. The discussion on how to learn compact embeddings of policies addresses a critical issue for applying PeVFA in practice.
* 2. The idea of taking the policy as an input to the value function approximator is interesting.

Things that can be improved:
* 1. I find Theorem 1 not particularly insightful - it basically says "when PeVFA generalizes well, it helps". Maybe the authors can make a more detailed comment on Theorem 1 in their response to help me understand its insight.
* 2. While the local generalization scenario is theoretically possible, it is unclear to me how it would work in practice when the PeVFA is implemented by standard deep neural networks. Intuitively, the policies along the policy improvement path have increasing values. Therefore the PeVFA must "extrapolate" to generalize well. However, it is known that standard deep neural networks are not very good at extrapolating without specialized architecture design. In the experiments, the PeVFAs were implemented by standard MLPs, which are not expected to generalize well. Yet they still demonstrated performance improvement. It is possible that my hypothetical argument is wrong, but I would find the empirical results more convincing if the authors can address this question.
* 3. I find the choice of implementing PeVFA with PPO rather intriguing. Based on Theorem 1, the larger the value difference between consecutive policies during policy iteration, the better the chance PeVFA could offer benefits via generalization. However, like other trust region policy optimization methods, PPO puts a constraint on consecutive policies so that the updated policy cannot deviate too far from the old policy. Intuitively PPO introduces an upper bound on the room for the improvement of PeVFA. Perhaps the authors could provide some explanation on why choosing PPO as the base agent.
* 4. It is unclear to me how the proposed policy embedding learning methods would scale up to larger policy networks. One possibility is to separate the policy network into a state embedding module and a policy head and only encoding the policy head which is usually smaller. Could the authors comment on how to scale up and perhaps provide some preliminary results?
* 5. L231-232, "... PeVFA can achieve lower approximation error than conventional VFA after the same number of training with the same on-policy samples". What does "the same number of training" mean here? Intuitively, PeVFA needs more training, though not necessarily more data, to outperform the conventional VFA because it is learning a more complicated function.

Minor errors and typos:
* 1. L301, delete "to"

**Time Spent Reviewing:**

4

---

> ### Author Response · Authors · 2021-08-10
> **Response to the Initial Review by Reviewer aYpG**
>
> We appreciate the reviewer’s careful and valuable comments.
>
> &nbsp;
>
> $\textbf{Q1:}$ More detailed comments on Theorem 1
>
> Please kindly refer to $\textbf{R1}$ of the $\textbf{Overall Response}$ for more remarks on Theorem 1, briefly including the motivation, the significance, more general cases.
>
> &nbsp;
>
> $\textbf{Q2:}$ More about how PeVFA generalizes in practice (Extrapolation of MLP and PPO as the base)
>
> We appreciate the reviewer for pointing out this interesting and important problem.
>
> The generalization and the policy improvement have an intriguing interplay between them in practice. This is also described by the sufficient condition presented in Theorem 1 since the generalization and the policy improvement lie on the two sides.
>
> For the generalization, we also consider that the ability of generalization (especially extrapolation) of conventional MLP networks is limited. This is also a major reason to our conservatism on the effectiveness of GTPI (discussed in Line 94-95 of Section 2) although PVN and PVFs have demonstrated the preliminary successes of GTPI in the environments like Cart Pole and MuJoCo Swimmer.
>
> We conjecture that the extrapolation can be poor when the policy changed a lot during the improvement path. This potentially reveals that PeVFA is well suitable to PPO due to its characteristics on ensuring both stable improvement and constrained policy change. This may also explain our empirical finding on that the improvement obtained by PeVFA for Advantage Actor-Critic (A2C) algorithm (between 10% - 20% for different environments) is less significant than PPO-PeVFA.
>
> Another reason we use PPO as the base is that PPO achieves stronger performance (closer to the SOTA) than A2C due to the more advanced policy improvement (optimization).
>
> Consider PPO narrows the RHS of the sufficient condition in Theorem 1, a complementary comment on the benefit of PeVFA is related to the more general sufficient conditions, as the third point we provided in $\textbf{R1}$ of the $\textbf{Overall Response}$. This may be somewhat easy to imagine when the extrapolation is assumed, since $\doubleV_{\theta_t}(\pi_t)$ is likely to be at the side of the directional extension line of the segment with a direction from  $V^{\pi_{t+1}}$ to $V^{\pi_t}$.
>
> We plan to investigate the fundamental problem (i.e., the interplay) in the future.
>
> &nbsp;
>
>
> $\textbf{Q3:}$ About how to scale up to large policy networks
>
> Since SPR does not directly depend on the scale of policy network, we assume the reviewer questions on OPR. In general, we may scale up to large policy networks from two aspects:
> 1) The first aspect is to reduce the policy parameters. One possible way is, as mentioned by the reviewer, to make use of the separation of state representation and a smaller policy head (e.g., even a linear policy layer) that takes as input of the former. This is intuitively reasonable while an apparent issue is the representation shift [1-2]. In other word, since the state representation is often learned along with the policies, an immediate consequence is that each policy head on the improvement path inherently depends on the state representations at that time. One may avoid this issue if pre-trained/pre-defined representations are available [3-4] or alleviate this issue by using a slow-update state representation and a sliding window for recent policies (though this may slow the practical learning). Several works study on such two-timescale networks [5] and we think more principled methods may be proposed in the future.
> 2) The second aspect is to propose better policy representations. We are aware of this and propose OPR that leverages PI transformations for abstract representations of layers. This allows OPR to be applicable to various MLP policies (though still be limited for other sophisticated architectures).
> To our knowledge, network (parameters) encoding is not well studied. Several works have also made very first attempts on this problem by naively making use of VAE latent representations [6] or the statistics of network layers [7]. This is also an important future direction of our work.
>
>  &nbsp;
>
>
> $\textbf{Q4:}$ Line 231-232 ‘the same number of training’
>
> We meant to say, to the current policy we want to evaluate, the same number of batches from the same on-policy samples (of the current policy) are used to train PeVFA. It is correct that the overall training cost of PeVFA is larger than the conventional VFA since we may need to train the PeVFA to approximate the values of historical policies.
>
> We appreciate the reviewer for pointing out this and we will clarify this to avoid the ambiguity.
>
> &nbsp;
>
> ---
>
> Reference:
>
> [1] The Impact of Non-stationarity on Generalisation in Deep Reinforcement Learning. Maximilian Igl, Gregory Farquhar, Jelena Luketina, Wendelin Boehmer, Shimon Whiteson. arXiv/2006.05826 (2020)
>
> [2] Implicit Under-Parameterization Inhibits Data-Efficient Deep Reinforcement Learning. Aviral Kumar, Rishabh Agarwal, Dibya Ghosh, Sergey Levine. ICLR 2021
>
> [3] Decoupling Representation Learning from Reinforcement Learning. Adam Stooke, Kimin Lee, Pieter Abbeel, Michael Laskin. ICML 2021
>
> [4] Reinforcement Learning with Prototypical Representations. Denis Yarats, Rob Fergus, Alessandro Lazaric, Lerrel Pinto. ICML 2021
>
> [5] Two-Timescale Networks for Nonlinear Value Function Approximation. Wesley Chung, Somjit Nath, Ajin Joseph, Martha White. ICLR 2019
>
> [6] Predicting Neural Network Accuracy from Weights. Thomas Unterthiner, Daniel Keysers, Sylvain Gelly, Olivier Bousquet, Ilya O. Tolstikhin. arXiv:2002.11448 (2020)
>
> [7] Policy manifold search: exploring the manifold hypothesis for diversity-based neuroevolution. Nemanja Rakicevic, Antoine Cully, Petar Kormushev. GECCO 2021

---

> > ### Comment · Reviewer_aYpG · 2021-09-02
> > **Thank you for the detailed response!**
> >
> > Thank you for the detailed response! Your response clarified my confusion about the empirical study.
> >
> > After discussing with other reviewers, I agree that the empirical section is fairly strong. However, as other reviewers pointed out as well, the paper is held back by the weak theory section and its disconnection to the empirical section. One possible solution is to drop the theory part entirely and focus solely on the empirical part, as suggested by reviewer RjEL. But the authors expressed their belief in the potential of the theoretical study and did not want to turn the paper into a purely empirical study, which I understand. Thus I would like to encourage the authors to refine their theory to improve the paper.
> >
> > Therefore, I do not think this paper is ready to publish in its current form. I will stick to my initial evaluation and will not increase my score.

---

### Official Review · Reviewer_NFfA · 2021-07-15

**Rating:** 5
**Confidence:** 3

**Summary:**

This paper studies a class of reinforcement learning methods which utilize a value function which in addition to the state is conditioned on some representation of the current policy. The idea being that this can allow the agent to learn a value function that explicitly generalizes among policies as the policy changes over the course of learning and ultimately improve performance. The approach is named Policy-extended Value Function Approximation (PeVFA).

They present some theoretical results with essentially show that if we assume that lowering the error in the value function for one policy results in sufficiently lower error for nearby policies, and the policy does not change too much between iterations, we can bound the value error for a policy dependent value function in terms of distance between policies at subsequent iterations. They also show that assuming the policy dependent value function generalizes well it can provide a benefit over a policy independent value function in terms of value error during a policy iteration procedure.

They further present empirical results using policy dependent value functions with a variety of different techniques for conditioning the value function on the policy. Policy representation techniques tested include: Raw Policy Representation (RPR) which uses all the parameters of the policy network as input to a value approximation, Surface Policy Representation (SPR) which uses a set of state action pairs gathered from running the policy, and Origin Policy Representation (OPR) which again uses the policy network parameters, but preprocessed by a permutation invariant transform. In addition, they test combining these representation strategies with auxiliary losses encouraging policy recovery from eh learned representation.

They study the use of policy dependent value functions in two settings, a supervised learning like setting where a number of fixed policies are initially evaluated to train the value function and a reinforcement learning setting where a policy is adapted online while learning a policy dependent value function for the policies the arise along the way. Results indicate that in the supervised case the learned value function can generalize to unseen policies.  In the reinforcement learning case, policy dependent value functions appear to confer some performance improvement when used with PPO compared to standard value functions.

**Limitations And Societal Impact:**

Limitations are reasonably well covered. I don't see significant potential for negative societal impact.

**Main Review:**

The basic idea of somehow learning policy conditioned value functions as a way to more rapidly adapt to a changing policy is interesting and reasonable. A key challenge in implementing this is obviously how to encode a policy in a way that can be input to a neural network. My overall impression is that the empirical aspect of this work was reasonable, but could be improved, while the theory does not seem terribly useful.

Having checked over the theory in reasonable detail the results appear correct to me. However, the theory is arguably not very useful as it's intuitively obvious that if we assume the value function generalizes between policies meaningfully such that reducing the loss for one policy reduces the loss for nearby policies sufficiently, then such a policy dependent value function function will confer a benefit over a policy independent one. The hard part is how to actually achieve such generalization, and how to encode a policy into a form that is convenient to input into a neural network.

Also Theorem 1, does not actually use any of the lemmas, which makes it strange to call them lemmas, and seems trivial. Essentially it just applies definitions and the triangle inequality to conclude that if the sum of the value errors for two subsequent policies is smaller then the actual difference between the true values of the policies, then the value error for the second policy will be less than the error in approximating the second policy by the value of the first. The authors claim that this theorem shows that we can get a benefit from using value functions that depend directly on the policy. However, it's clear that such a benefit is possible if the policy dependent value function happens to generalize well even without this theorem, and it's not really clear to me what this theorem adds. Again the hard part is actually ensuring such favourable generalization occurs.

On the empirical side the results for learning the value of multiple pertrained policies are interesting and seem to demonstrate the possibility of improving generalization to new policies using PeVFA. The proposed algorithm for reinforcement learning seems to rely on growing a buffer containing trajectory data for all past policies and repeatedly retraining on it, which seems fairly restrictive in practice. Nonetheless, it is perhaps a reasonable way to start in order to highlight the potential utility of PeVFA. The results seem to demonstrate the PeVFA can provide an advantage over standard VFA in the settings explored, however aspects of the empirical evaluation could be improved and clarified to make the conclusion more clear.

According to the appendix, the added hyperparameters for PPO-PeVFA were tuned using a grid search, but the hyperparameters of the PPO baseline were not which perhaps confers an unfair advantage to PPO-PeVFA. Furthermore, I couldn't find any details on how the shared hyperparameters were selected this should be stated clearly to aid in interpreting the results.


Minor Corrections and Suggestions
=================================
Line 34 'efficiency value approximation'->efficiency of value approximation'

Lemma 1: Is the assumption $f(\pi)_1\leq f(\pi_2)$ really necessary here? It seems a little unnatural and it seems to me you could drop it and modify the result to $\hat{f}(\pi_2)\leq\gamma f(\pi_2)+\gamma \mathcal{M}(\pi_2,\pi_1,\hat{L})+\mathcal{M}(\pi_1,\pi_2,\hat{L})$

For proofs in the appendix: it would help readability if you rewrite the theorems/lemmas before the proof.


Questions for authors
=====================
Line 313: What is meant by Ran PR? I can't find any details on this and I don't really know what is meant by "randomly generated policy representation". Randomized how exactly?

It seems quite odd to me that OPR uses permutation invariant transformations. Surely the ordering of the weights and biases is important for a multilayer network, what is the justification for processing it in this way?

**Time Spent Reviewing:**

10

---

> ### Author Response · Authors · 2021-08-10
> **Response to the Initial Review by Reviewer NFfA**
>
> We appreciate the reviewer’s careful and valuable comments.
>
> &nbsp;
>
> $\textbf{Q1:}$ Motivation and Focus of the Theory
>
> The benefit of PeVFA is intuitively obvious for the global generalization case which can be viewed as a typical supervised learning problem. Our Lemma 1 and Corollary 1 describes the effective generalization (i.e., the generalized contraction) by general and simple formulations.
>
> We note that the situation is different for typical RL under the GPI paradigm. The effective generalization (intuitively generalize well) is not sufficient to ensure the benefits of PeVFA along the GPI path. This is because the approximate values for the predecessor policy (i.e., without value generalization among policies) may be closer to the true values of current policy which are to approximate next. As a consequence, the conventional VFA can be preferred than PeVFA in such cases.
>
> Therefore, we provide Theorem 1 to present a sufficient condition (actually the strictest) to the case where PeVFA can benefit the process of typical RL. The LHS of the sufficient condition presented in Theorem 1 can be analyzed by the basic generalization behaviors provided in Lemma 1-2. This also sheds some light on the rationality of the presented condition. We will amend our theory to make more proper use of lemmas and theorems as pointed out by the reviewer.
>
> We refer the reviewer to $\textbf{R1}$ of the $\textbf{Overall Response}$ for some additional remarks on our theory.
>
> &nbsp;
>
> $\textbf{Q2:}$ Growing buffer and retraining
>
> A growing buffer (with maximum in FIFO fashion) for transitions or trajectories is commonly used by off-policy RL algorithms. We do not consider our algorithm adds severe extra requirements of memory: for SPR, only a policy index is added to each state-action pair (or trajectory); for OPR, the parameters of each policy network encountered is stored.
> Note that the policies are usually 2-3 orders of magnitude lesser than transitions for on-policy algorithms (if count in iteration and neglect the intermediates of each single update). For off-policy algorithms where policies are often updated for once at a small interval of several timesteps, the policies can be stored with a proper frequency. In addition, we may not have to care about the policies that are out of date a lot.
>
> For retraining, in our experiments, 100 batches of 64 state and MC return pairs is conducted for PeVFA training every 1000 timesteps (i.e., interactions), which is comparable (slightly light-weighted) to the convention VFA training, i.e., $1000 / 128 * 10 \sim 78$ batches of 128 state and MC return pairs every 1000 timesteps. For representation training, the scheme is also commonly adopted in the works on incorporating like state representation learning into RL. In our algorithm, policy representation training is conducted at lower frequency with smaller batch sizes.
>
> Empirically, our PPO-PeVFA variants almost take less than 2x the wall-clock time of PPO. It also remains much space for the improvements on these training schemes.
>
> &nbsp;
>
> $\textbf{Q3:}$ Hyperparameters of PPO and PPO-PeVFA
>
> We manually tune our standard PPO baseline by referring to the implementations in a few previous works. Note that one can refer to the results of standard PPO (with Tensorflow implementation) in the MuJoCo benchmarks of OpenAI Spinning Up (https://spinningup.openai.com/en/latest/spinningup/bench.html) for a calibration. The results of our PPO are comparable (or even better) than those reported by OpenAI Spinning Up benchmarks.
>
> We then build PPO-PeVFA on our PPO implementation. The same parameters are used for the common part of PPO and PPO-PeVFA (see Table 2 in Appendix) and only the hyperparameters for extra components of PPO-PeVFA is tuned as shown in Table 3 (actually in a small range of grid search).
>
>
> &nbsp;
>
> $\textbf{Q4:}$ Random policy representation (i.e., Ran PR)
>
> For Ran PR, a random representation vector whose value of each dimension is sampled from the uniform distribution $U(-1,1)$, is used for each policy. Thanks for pointing out this and we will clarify this as suggested.
>
> &nbsp;
>
> $\textbf{Q5:}$ Permutation invariant (PI) transformations for OPR
>
> The most straightforward method to encode a policy network is to vectorize the parameters and feed it into an MLP. However, since the parameter vector is high-dimensional, this may require a deep MLP (or other sophisticated architectures) with plenty of parameters and strong update signals (e.g., with large number of diverse policies) to make the learned low-dimensional policy representation work. In addition, such a method does not allow us to encode the policy networks with different numbers of units by a single encoder network.
> The main reason why we use PI transformations for OPR encoding is to reduce the difficulty of representation learning as mentioned above, and meanwhile to make the representation applicable to various policy MLPs with different number of layer units (e.g., 64 ---> 512). A nature idea is to use PI transformations which are often adopted to deal with sequential data.
>
> Moreover, PI transformations are performed for each layer separately. Consider the parameters $\{(w_i, b_i)_{i=1^{h}}\}$ in each layer, where $h$ is the number of units in this layer, $w_i$ is a $h^{\prime}$-d vector ($h^{\prime}$ is the number of units in the previous layer) and $b_i$ is a scalar. Each parameter pair $(w_i, b_i)$ defines an operation on the features outputted by the previous layer. Thus, we intuitively perform PI transformations for these parallel operations to obtain a layer representation. Such a layer representation can be viewed as the generalization of the layer statistics proposed in [1] for supervised classifier networks. We are aware of the limitation of our OPR encoding (partially discussed in Line 813-816 in Appendix).
>
> To our knowledge, network (parameters) encoding is not well studied. Several works naively make use of VAE latent representations [2] for very first attempts on this problem. This is also an important future direction of our work.
>
> &nbsp;
>
> $\textbf{Q6:}$ Minors
>
> In Lemma 1, Lipschitz continuity is only assumed for $f_{\hat{\theta}}$ with constant $\hat{L}$. To drop the condition $f_{\theta}(\pi_1) \le f_{\theta}(\pi_2)$ and derive the bound as pointed out, an additional assumption of  Lipschitz continuity on $f_{\theta}$ with the same constant $\hat{L}$ is needed. We discuss the condition $f_{\theta}(\pi_1) \le f_{\theta}(\pi_2)$ in Remark 1 (Line 580).
> We thank the authors for other valuable suggestions.
>
> &nbsp;
>
> ---
>
> References:
>
> [1] Predicting Neural Network Accuracy from Weights. Thomas Unterthiner, Daniel Keysers, Sylvain Gelly, Olivier Bousquet, Ilya O. Tolstikhin. arXiv:2002.11448 (2020)
>
> [2] Policy manifold search: exploring the manifold hypothesis for diversity-based neuroevolution. Nemanja Rakicevic, Antoine Cully, Petar Kormushev. GECCO 2021

---

> > ### Comment · Reviewer_NFfA · 2021-09-01
> > **Thank you for the response**
> >
> > I thank the authors for their response. After reading through the author's responses and discussion with the other reviewers my opinion remains mostly the same.
> >
> > I feel that my empirical concerns have been fairly well addressed, in particular regarding hyperparameter tuning. I feel the described methodology is reasonable and would encourage the authors to add these details to the paper.
> >
> > I appreciate the expanded motivation for the theory, however, I still feel like the current theoretical component of the paper provides fairly limited additional value to the reader in understanding or motivating the method.

---

> > > ### Author Response · Authors · 2021-09-02
> > > **We appreciate the reviewer's response**
> > >
> > > We appreciate the reviewer's response very much! Many thanks to your careful review again. All your reconstructive suggestions will be taken to improve our work in the future.

---

### Official Review · Reviewer_RjEL · 2021-07-16

**Rating:** 4
**Confidence:** 5

**Summary:**

The authors propose PeVFA, an extension to value functions that include not only states (or state-action pairs), but a representation of the parameters of the policy. They apply their method to PPO and show experimentally that such policy representations help improving over standard PPO. Unfortunately, the theoretical results in the paper seem trivial and disconnected from the rest of the paper. The value functions presented in the paper were already introduced in previous work. The authors compare extensively these methods to their approach, but introduce their PeVFAs separately, creating some confusion on the novelty of the work.


**Limitations And Societal Impact:**

Yes

**Main Review:**



  - I like the extensive comparison between PeVFAs and PBVFs[1]/PENs. I think this work can provide and important contribution in that line of research by providing useful ways of representing the policy parameters. This tackles the main limitations of PBVFs. However, I think the authors should present their work as an improvement over PBVFs (in particular the PSVF), rather than introducing PeVFAs from scratch and then claiming after that this was already done in PBVFs. Note that PBVFs are theoretically sound to receive any representation of the policy, although in their experiments no representation is used.
   - The theory proposed is a basic analysis result and should be omitted. In particular, if a function is L-continuous and it decreases in a point, then there exists another point close enough such that the function decreases also there. Moreover, the contraction property is never holding in the experiments performed, hence the theoretical results are completely disconnected from the paper.
  - The assumption in Theorem 1 seems to be a technical step that is needed to prove the theorem itself, but not a reasonable assumption. It states that the sum of the losses of two consecutive policies has to be lower than the distance between the value functions of the two policies. How can one interpret this?
 - In line 229 the authors claim that Figure 3 demonstrates the consequences of Theorem 1. This is not possible, since the assumptions of Theorem 1 are never met.
- I think the term GPTI is never defined
- PBVFs are used for offline learning (not online)
- the citation on PBVFs refers to an old ArXiV submission and not to the version published [1]


Again, the contribution relative to policy representation seems interesting and useful and it should be the main focus, but the paper in its present form is not ready for acceptance. If the authors remove the theoretical part and address my concerns about how they introduce PeVFAs, I will increase the score accordingly.

[1] Francesco Faccio, Louis Kirsch, and J¨urgen Schmidhuber. Parameter-based value functions. In Int. Conf. on Learning Representations (ICLR), Virtual only, May 2021


**Time Spent Reviewing:**

5

---

> ### Author Response · Authors · 2021-08-10
> **Response to the Initial Review by Reviewer RjEL**
>
> We appreciate the reviewer’s careful and valuable comments.
>
> &nbsp;
>
> $\textbf{Q1:}$ About Policy Evaluation Network (called PVN in their original paper [1], or PEN called by the reviewer), PVFs (also called PBVFs in their public version [2]) and PeVFA
>
> In fact, we are not aware of the works of PVN and PVFs until we completed our first version of this paper. The discussions on PVN and PVFs are then added in Section 2 (as well as some content in our experiments and appendix). As revealed by our discussions, we agree with reviewer that one can view PeVFA as a generalization of PVFs since PeVFA considers the policy representations (embeddings) instead of the original policy parameters as inputs.
>
> Another thing is that PSSVF $V(\theta)$ in PVFs can be viewed as a generalization of PVN since PSSVF replaces Network Fingerprint by vectorized policy network parameters and provides an online learning algorithm. See Section 5 of [2] for more on such a generalization.
>
> After all the above, we will consider to accept the reviewer’s suggestion if the reviewer insists that our work should be proposed as an improvement of PVFs rather than from scratch.
>
> &nbsp;
>
> $\textbf{Q2:}$ Theory
>
> Please kindly refer to $\textbf{R1}$ of the $\textbf{Overall Response}$ for some additional clarifications.
>
> We also appreciate the review for providing his/her concerns on our theoretical results. We will consider to add some deeper theoretical analysis (as partially mentioned in $\textbf{R1}$) and remove some less important content as suggested.
>
> &nbsp;
>
> $\textbf{Q3:}$ For ‘PBVFs are used for offline learning (not online)’
>
> In Line 103-104 in our Section 2, we note that PVFs are proposed for both online and offline learning in the original paper (see their Algorithms and experiments in their Section 4.2).
>
> &nbsp;
>
> Thanks for pointing out the published version. We will replace the old reference by the public one.
>
> ---
>
> Reference:
>
> [1] Policy Evaluation Networks. Jean Harb, Tom Schaul, Doina Precup, Pierre-Luc Bacon. arXiv:2002.11833 (2020) https://arxiv.org/abs/2002.11833
>
> [2] Francesco Faccio, Louis Kirsch, and J¨urgen Schmidhuber. Parameter-based value functions. In Int. Conf. on Learning Representations (ICLR), Virtual only, May 2021. https://openreview.net/forum?id=tV6oBfuyLTQ

---

> > ### Comment · Reviewer_RjEL · 2021-08-26
> > **Response to the authors**
> >
> > I thank the authors for their feedback.
> >
> > >After all the above, we will consider to accept the reviewer’s suggestion if the reviewer insists that our work should be proposed as an improvement of PVFs rather than from scratch.
> >
> > The authors should at least acknowledge in line 128-130 that the functions $V(s,\theta)$, $Q(s,a,\theta)$ were first introduced in the PBVF paper. In the current version of the paper, only the PVN/PSSVF $V(\theta)$ is mentioned. Can the authors please further elaborate on this request?
> >
> > >In Line 103-104 in our Section 2, we note that PVFs are proposed for both online and offline learning in the original paper (see their Algorithms and experiments in their Section 4.2).
> >
> > I thank the authors for clarifying this. When not used for full offline learning (like I see in their Sections 4.3 and 4.4) PBVFs alternate online steps in which a new policy is acting in the environment with offline optimization steps where the value function and policy are learned offline, from the replay buffer. Of course, the PSVF and PAVF can be learned fully online, optimizing after every time step in the environment, but this is not what happens in their experiments. This is why I was not sure what the authors meant for "online policy learning" in line 104.
> >
> > I agree with the authors on the fact that a comparison between PPO-PeVFA and PPO (like in the current version of the paper) is more meaningful than a direct comparison between PPO-PeVFA and PVNs/PBVFs (although I would have liked to see the fingerprint mechanism of PVNs applied to PPO-PeVFA). I think that the current experimental results are already convincing and show the benefits of the proposed representations when applied to PPO. One thing I would like to see as future work is an application of the proposed dimensionality reduction techniques to PBVFs and a comparison with PBVFs that use no policy representation.
> >
> > After reading the discussion from the authors and from the reviewers, I am still convinced that the theory proposed in the current version of the paper is rather trivial and does not provide useful insights. I would like to ask the authors if they would consider removing it completely from the paper and from the Appendix (in particular Def. 1,2, Lemma 1, 2, Corollary 1,2, Theorem 1), in favor of a more detailed explanation of the proposed dimensionality reduction techniques

---

> > > ### Author Response · Authors · 2021-08-27
> > > **Response to the Response #1 by Reviewer RjEL**
> > >
> > > We appreciate the reviewer's feedback.
> > >
> > > &nbsp;
> > >
> > > $\textbf{R1}:$ About $V(s, \theta)$ and $Q(s,a,\theta)$
> > >
> > > We acknowledge that $V(s, \theta)$ and $Q(s,a,\theta)$ are first studied in PBVFs, but $\textbf{for the first introduction (or mentioned)}$ of $V(s, \theta)$ and $Q(s,a,\theta)$, we may consider it $\textbf{should be in PVN}$ (see the second paragraph of their Section 8), which is publicly available at Feb 2020 (4 months earlier than the first public version of PBVFs). This is also why we may view PBVFs as a development of PVN. Of course, the authors of PBVFs may not be aware of the PVN during their work.
> > >
> > > Our work focuses on a $\doubleV(s, \chi_{\pi})$ (and $\doubleQ(s,a, \chi_{\pi})$) from the begining (as classified in the initial response), and later we found the works of PVN and PBVFs. We classify this here to help the reviewers know the relations among them. Our work is concurrent to PVN and PBVFs, although this point may be not important.
> > >
> > > &nbsp;
> > >
> > > $\textbf{R2}:$ Online and Offline Leanring of PBVFs
> > >
> > > Actually, as in their Section 4.2, online learning is what $\textbf{exactly}$ happens in their experiments, just following the illustration in their algorithms (Algorithm 1 to 4. i.e., interact, train PBVFs, optimize policy and repeat).
> > >
> > > This can also be verified in their opensource code (somewhat surprising, the code is uploaded to github just $\textbf{two days after}$ I posted the $\textbf{Overall Response}$ above!). By the way, their opensource code also only contains the part for online learning of PBVFs at present (although other experiments in their experiments can be reproduced with their online learning code).
> > >
> > > &nbsp;
> > >
> > > $\textbf{R3}:$ '... removing the theory completely ...'
> > >
> > > As classified in the $\textbf{Overall Response}$, we consider that our theory is related to two problems: $a)$ how are the values generalized among policies, $b)$ what are such cases of generalization like and what are the exact sufficient conditions to achieve them, especially under the GPI paradigm. We emphasize the importance of the two problems which is also not studied in previous works.
> > >
> > > To our work, this is also an important support to the effectiveness of the local generalization along the improvement path. Besides, our theory occupies near 1 of 9 pages.
> > >
> > > Due to the reasons mentioned above, we tend to improve our theory (further from current preliminary results) intead of removing it completely (which will cripple the rationality of proposing GPI with PeVFA).
> > >
> > >
> > > &nbsp;
> > >
> > > We appreciate the reviewer's other suggestions which will be considered to improve our work in the future.

---

> ### Author Response · Authors · 2021-08-30
> **A furthe discussion**
>
> > In line 229 the authors claim that Figure 3 demonstrates the consequences of Theorem 1. This is not possible, since the assumptions of Theorem 1 are never met.
>
> Can the reviewer further explain this kindly, i.e., how to interpret the 'not possible' and 'never met'. It will be valuable to us for some deeper understandings and potential inspirations from different angles.
>
> Thanks!

---

> > ### Comment · Reviewer_RjEL · 2021-09-02
> > **About Theorem 1**
> >
> > In theorem 1 the authors prove a statement "A implies B". The fact that in Figure 3 we observe B does not tell anything about the existence of A or the implication itself. If the authors want to claim that Figure 3 demonstrates the consequences of Theorem 1, then they should first show that A is holding.
> >
> > Perhaps a simple experiment compatible with the theory could be the following:
> > - Consider a finite-state MDP and a dynamic programming algorithm with PeVFA where the contraction property holds.
> > - Show that the (other) assumption of Theorem 1 holds in this setting.
> > - Then run the algorithm and show that also the result implied by Theorem 1 holds.
> >
> > I think that this paper without theory could be ready for publication, but the theory in its present form is very disconnected from the paper. If the authors prefer to improve the theoretical section, I strongly suggest showing how the assumptions of Theorem 1 are holding in a simple setting before scaling up their experiments with PPO.
> >
> > Nevertheless, I still think the authors might want to remove some of the theoretical results. It is obvious that if a function is smooth and decreases in a point, then there exists another point close enough such that the function decreases also there

---

> > > ### Author Response · Authors · 2021-09-03
> > > **We Appreciate the Reviewer's Concrete Suggestion. And Other Discussions**
> > >
> > > Thanks for the reviewer's valuable feedback!
> > >
> > > &nbsp;
> > >
> > > We agree that we observe B in Figure 3 and we cannot claim the existence of A.
> > > We are aware of this and thus in fact, we say 'demonstraing the $\text{consequence arrived in Theorem 1}$' (Line 228-230), i.e., the results shown in Figure 3 demonstrate the existence of B.
> > >
> > > This may be a place that is easy to be misunderstood. We will clarify this to remove the ambiguity.
> > >
> > > &nbsp;
> > >
> > > We appreciate the reviewer's concrete suggestion about the finite MDP experiments. This is valuable to us and we will carefully consider it.
> > >
> > > &nbsp;
> > >
> > > We thank the reviewer's recognition on our work expect for the theory. This is an insipiring support.
> > >
> > > One of our concerns about removing the theory is that, although it is indeed straightforward for the generalization of a smooth function, one other reviewer may ask $\text{'we agree that PeVFA can generalize but why is it necessarily better than the conventional VFAs?'}$.
> > > To this end, we thought we have to describe the scenario and at least provide some potential situations, as we do in the submitted version.
> > >
> > > &nbsp;
> > >
> > > Finally, to be honest, we do not clearly see why 'assumptions of Theorem 1 are never met'. We conjecture the reviewer means the assumptions are never validated empirically in a direct way in our work.
> > >
> > > In this work, we are intended to say that $B$ observed in Figure 3 implies a possibility for A
> > > to be met (actually, there should be many less strict assumptions to imply B, as mentioned in the $\textbf{Overall Response}$).

---

> > > ### Author Response · Authors · 2021-09-04
> > > **We are Happy to Accept Reviewer's Suggestion on Paper Amendment**
> > >
> > > We are happy to accept reviewer's suggestion to amend our paper (1 of 9 pages) and remove our priliminary (and simple) theory.
> > >
> > > To make the paper self-contain and the logic chain complete, we plan to replace the theory with more intuitive and descriptive contents to render the local generalization scenario and the rationality of the generalization for GPI.
> > >
> > > We appreciate the reviewer's recognition on our work expect for the theory again.

---

### Official Review · Reviewer_KgSY · 2021-07-16

**Rating:** 7
**Confidence:** 3

**Summary:**

Update: See the comments. Review score remains at 7.

The paper considers learning value function approximators that take a policy representation as an input to the value function to allow generalization across different policies. There are several other recent works of this type. The aspect differentiating the current work is the type of policy representation used that allowed the method to scale to policies with 10^5 parameters.

The paper considered two types of data sources for the policy representations:
1. State action pairs obtained by the policy.
2. The policy parameters.

Then, the policy representation is formed by passing the data through a multilayer neural network that embeds the data into a lower dimensional representation.
The embedding is fed into the value function as an input, and the network creating the embedding is trained end-to-end to minimize the value function prediction error. Optionally they also added auxiliary losses such as contrastive learning.

Experiments were performed using PPO combined with their new value function and compared to the standard PPO as well as variants taking a random policy representation and also a policy representation with no emebedding. They performed experiments on 6 MuJoCo tasks, and generally the new methods outperformed the baselines.

tSNE plots showed that the policies with similar representations had a similar performance at the task (when comparing the achieved rewards). Moreover, experiments showed that the value prediction error was smaller for the new method compared to a standard value function method, and also that the value function was accurate for a test set of new policies not encountered during training.

**Ethical Concerns:**

No concern.

**Limitations And Societal Impact:**

Yes.

**Main Review:**

The paper was reasonably well written. The experimental work examined the performance from multiple aspects and showed that their method could deal with the issue of the large parameter space of the policy by embedding it to a smaller space.

I have a question about the state action pair embedding: When you are applying the value function to a new policy, do you first need to gather new data to perform the embedding to compute the value?

One down-side of the work is that it is not compared to the other recent works doing a similar thing, e.g.
J. Harb, T. Schaul, D. Precup, and P. Bacon. "Policy evaluation networks"
or
F. Faccio and J. Schmidhuber. Parameter-based value functions.

Although, these works are quite recent, so maybe it's not expected.
The paper could also benefit from fixing some English language errors, but it's not a big issue.
I would also suggest changing the title: I'm particular confused by why the "Represent your own policies" is added in the beginning. Doesn't everyone decide their own representations?

**Time Spent Reviewing:**

3

---

> ### Author Response · Authors · 2021-08-10
> **Response to the Initial Review by Reviewer KgSY**
>
> We appreciate the reviewer’s careful and valuable comments.
>
> &nbsp;
>
> $\textbf{Q1:}$ State-action pair embedding
>
> As in the line 3-4 of Algorithm 1 (as well as the Line 4-5 in Algorithm 3), a new policy $\pi$ after the update in the last iteration collects on-policy interaction experiences, which are stored and used to calculate the policy representation of $\pi$ (i.e., SPR). This scheme usually suits on-policy RL algorithms well and is exactly how we implement our algorithms (PPO-PeVFA) in our experiments.
>
> In principle, on-policy experiences (or new data of new policy) are not necessarily needed. Since we can often have a buffer of states which are obtained by some previous policies or others, we can generate actions for these states by sampling from any new policy $\pi$. In this way, we can obtain the state-action pairs without interacting with the environment. For this more general scheme, the selection of the off-policy state distribution to depend on can be a question to consider. We regard this more general case as future work.
>
> &nbsp;
>
> $\textbf{Q2:}$ Comparison to PVN and PVFs
>
> For the aspect of algorithm, we introduce PVN and PVFs in Section 2 (start from Line 90) and also provide the differences of our work.
> For the empirical comparison, please refer to the $\textbf{R2}$ of the $\textbf{Overall Response}$ for the clarification, briefly including the inapplicability of PVN (offline v.s. online), the missing of official opensource implementations of PVN and PVFs and our failure in fully reproducing the results of PVFs.
>
> &nbsp;
>
> Other valuable suggestions (e.g., changing the title) will be considered.

---

> > ### Comment · Reviewer_KgSY · 2021-09-02
> > **Seems sensible**
> >
> > Thank you, that seems sensible.
> >
> > The other reviewers discussed the theoretical sections a lot. My positive assessment is primarily based on the empirical sections of the work, and I agree with the other reviewers that in its current state, the benefit of the theoretical sections may be limited. In particular the condition in Theorem 1 makes it not very informative. As the authors have said, they could extend the theory by analyzing the condition based on the other lemmas in the theoretical section. But this may be too much of a revision to allow accepting at this stage. I also wonder whether extending the theoretical section may make the contents too long for explaining clearly in a single paper.
> >
> > Nevertheless, I'll keep my score because I believe the current contents are good enough. In particular, the empirical evidence was convincing in showing that the method works for learning an embedding of the policies, and improving the accuracy of the value function.

---

> > > ### Author Response · Authors · 2021-09-02
> > > **We Appreciate the Reviewer's Support**
> > >
> > > We appreciate the reviewer's support very much! The length of the paper is indeed one of our concerns about this paper. But as mentioned by other reviewers, our theory is priliminary and there must be at least one better way to improve the theory and our current work. We will carefully consider this and try to find such a way in the future.
> > >
> > > Thank you very much again for your careful comments and your inspiring support and recognization.

---

> > > > ### Comment · Reviewer_KgSY · 2021-09-02
> > > > **Another option**
> > > >
> > > > You may also want to consider just writing a separate paper with an extended version of the theory, while removing the theory from the current paper.

---

> > > > > ### Author Response · Authors · 2021-09-02
> > > > > **Thanks for the Suggestion**
> > > > >
> > > > > Thanks for the suggestion! We will consider it, as also mentioned by Reviewer RjEL.
> > > > >
> > > > > Anthor one of our concerns is that, if we remove our theory (currently 1 of 9 pages), the logic chain can be incomplete. It is very likely that some reviewer may ask 'why the local generalization along the policy improvement path can be beneficial', 'why PeVFA is necessarily better than the conventional VFAs', 'what the situation can be for the occurrence of the former consequences', and etc. Therefore, we hope our theory to illustrate the potential situation and to provide at least some priliminary insights.
> > > > >
> > > > > Anyway, we will carefully consider both the options. Thanks for the suggestions by all the reviewers.

---

### Author Response · Authors · 2021-08-10
**An Overall Response about Additional Clarifications on the Theory and Experiments**

We appreciate the reviewers’ careful and valuable comments. We provide an overall response to the main concerns below and we provide individual responses for other questions later.

We also look forward to further discussions in the next reviewing phase to address the reviewer’s concerns if possible.

&nbsp;


$\textbf{R1: Additional Remarks on Our Theoretical Results}$

We appreciate the reviewers’ valuable comments which inspire us to improve our theory.
1) $\textbf{Motivation:}$
We agree it is intuitively apparent that an RL agent can benefit from PeVFA if the values can be well generalized among policies (i.e., so called $\texttt{meaningfully}$ as mentioned by $\textbf{Reviewer NFfA}$). This is also the thing often taken for granted. Our theoretical results are to provide some preliminary studies on: $a)$ how are the values generalized among policies and $b)$ what are such cases of $\texttt{meaningful}$ generalization like and what are the exact sufficient conditions to achieve them, especially under the GPI paradigm.

2) $\textbf{Theorical Results:}$
Our Lemma 1-2 study the question $a)$ by describing the generalization behavior based on general and simple formulations. Further, one important thing is that the generalized contraction (intuitively generalize well) is not sufficient to ensure the benefits of PeVFA for RL when considering the GPI path. This is because the approximate values for the predecessor policy (i.e., without value generalization among policies) may be closer to the true values of current policy which are to approximate next. As a consequence, the conventional VFA can be preferred than PeVFA in such cases.
Thus, our Theorem 1 (for question $b)$) presents a sufficient condition to the case where PeVFA can benefit the process of typical RL. For the sufficient condition $f_{\theta_t}(\pi_t) + f_{\theta_t}(\pi_{t+1}) \le \|V^{\pi_t} – V^{\pi_{t+1}}\|$ in Theorem 1, the LHS and the RHS are related to the value generalization and the policy improvement respectively. Taking the RHS as a fixed bound, the LHS is likely to be small enough given effective generalization and sufficient training.
For deeper looks on this, we note that it may involve the basic generalization provided in Lemma 1-2, other specific assumptions on generalization (as for the extrapolation mentioned by $\textbf{Reviewer aYpG}$), the policy improvement (e.g., the interplay between the approximation error, policy improvement and local generalization), as mentioned in our future work discussion.

3) $\textbf{More on Theorem 1}$:
In fact, we conjecture that there are many sufficient conditions that lead to the consequence of Theorem 1, and the presented condition (i.e., $f_{\theta_t}(\pi_t) + f_{\theta_t}(\pi_{t+1}) \le \|V^{\pi_t} – V^{\pi_{t+1}}\|$) is the strictest one among them to achieve. This can be interpreted by considering the geometrical relationship between $V^{\pi_t}$, $V^{\pi_{t+1}}$, $\doubleV_{\theta_t}(\pi_t)$ and $\doubleV_{\theta_t}(\pi_{t+1})$. One special case that requires the sufficient condition presented in Theorem 1 is where $\doubleV_{\theta_t}(\pi_t)$ and $\doubleV_{\theta_t}(\pi_{t+1})$ lie on the line segment between  $V^{\pi_t}$ and $V^{\pi_{t+1}}$.
We plan to investigate such conditions in a more complete fashion in the future.

&nbsp;


$\textbf{R2: Additional Clarifications on Being Unable to Provide Empirical Comparison}$
$\textbf{with PVN [1] and PVFs [2]}$

(P.S.: Policy Evaluation Network is called as PVN in their original paper [1] (or PEN called by $\textbf{Reviewer RjEL}$).
Parameter-based Value Functions are called as PVFs in their arXiv versions (also called as PBVFs in their public version on ICLR 2021 [2]). We use the abbreviations PVN and PVFs in our responses.)

In addition to the discussion in Line 90-114 in our paper, we provide some more clarification on this below (also as the response to the corresponding concern raised by $\textbf{Reviewer KgSY}$):
1) For PVN, it is proposed for a zero-shot (i.e., offline) policy optimization problem among Cart Pole and MuJoCo Swimmer. In principle, the policy representation called network fingerprint used by PVN can be evaluated separately; but we do not know how the exact implementation should due to the lack of their official code and we are also a little worried about the difficulty of optimizing the set of probing states in the experimental environments we considered. Besides, network fingerprint is showed to be not necessary in the experiments of [2] (see the discussion in their Section 5).
For the possible online variant of PVN, we consider this is partially done by [2], since PSSVF can be viewed as such a generalized variant. PSSVF derives an online learning algorithm (see their Algorithm 1) and is also empirically evaluated in their Section 4.2 and Figure 2.
2) As to PVFs (i.e., PBVFs), including PSSVF, PSVF and PAVF, we are not able to obtain the official implementations from the authors till now. The ICLR 2021 public paper says ‘Code is available at: https://github.com/FF93/Parameter-based-Value-Functions’ but the code repository is empty there (created on 18 March 2021). We also tried a few times to email the authors but did not receive a response. Moreover, we made a few attempts to reproduce their results on our own and we succeeded in reproducing the results of PSSVF (their Figure 2) in their environments under the online setting. Unfortunately, our reproduced PSSVF failed in the environments in our experiments, and currently we have not reproduced the results of PSVF and PAVF yet. Therefore, we conjecture this may be due to our personal reproduction. We can provide our reproduced implementations if needed.
For the separate evaluation of using vectorize policy network parameters as the representation, we provide the results of Raw Policy Representation (RPR) in our experiments.




---

Reference:

[1] Policy Evaluation Networks. Jean Harb, Tom Schaul, Doina Precup, Pierre-Luc Bacon. arXiv:2002.11833 (2020) https://arxiv.org/abs/2002.11833

[2] Francesco Faccio, Louis Kirsch, and J¨urgen Schmidhuber. Parameter-based value functions. In Int. Conf. on Learning Representations (ICLR), Virtual only, May 2021. https://openreview.net/forum?id=tV6oBfuyLTQ

---

### Decision · Program_Chairs · 2021-09-27

**Decision:**

Reject

**Comment:**

The reviewers agree that the paper tackles an important problem and that the proposed approach has its merits. They also agree that the paper is clear and easy to follow for the most part. The main concern expressed in the reviews and discussions regards the theory, which was considered rather trivial and somewhat disconnected from the rest of the paper. The overall opinion is that the main result builds on fairly strong assumptions that render them somewhat vacuous:  it is intuitively obvious that if the proposed approximator generalizes well across policies this should be useful. The real challenge lies in how to enforce this property in the first place.

Given the above, the reviewers suggested during the discussion that the authors remove the theory and present the paper as essentially an empirical argument. Although the authors initially resisted the idea, eventually they agreed to adopt the suggestion. There was some discussion whether the paper without the theory should be accepted or not, as all the reviewers seem to agree that the empirical part of the paper is reasonably well executed and has some interesting insights in it.

We eventually decided to reject the submission for two main reasons. First, the removal of the theory would be a fairly extensive modification to the paper. Second, and more important, it is unclear how much of a contribution the resulting empirical study would be with respect to the literature. The idea of adding an extra dimension to the value function corresponding to policies is quite straightforward and has appeared in the literature multiple times in different forms. Since the idea is already out there, the main challenge that remains to be resolved is how to represent these policies in a way that clearly promotes generalization across policies –especially those policies along the value-improvement path.

This work can be seen as a promising first step towards the goal above, but more work must be done in this direction. We hope the feedback provided can help in defining the next steps of this research.